# Enhanced Lung Cancer Classification Accuracy via Hybrid Sensor Integration and Optimized Fuzzy Logic-Based Electronic Nose

**DOI:** 10.3390/s25175271

**Published:** 2025-08-24

**Authors:** Umit Ozsandikcioglu, Ayten Atasoy, Selda Guney

**Affiliations:** 1Department of Electrical and Electronics, Faculty of Engineering, Karadeniz Technical University, 61080 Trabzon, Türkiye; ayten@ktu.edu.tr; 2Department of Electrical and Electronics Engineering, Baskent University, 06790 Ankara, Türkiye; seldaguney@baskent.edu.tr

**Keywords:** breath analysis, hybrid sensor-based electronic nose, lung cancer detection, data classification and fuzzy logic algorithm, nature-inspired optimization algorithms

## Abstract

**Highlights:**

**What are the main findings?**
A hybrid electronic nose system integrating 8 MOS and 14 QCM sensors effectively distinguished between lung cancer patients and healthy individuals through breath analysis.The fuzzy logic classifier optimized by a nature-inspired algorithm outperformed traditional methods, achieving 97.93% accuracy.

**What are the implications of the main findings?**
Demonstrates the strong potential of noninvasive electronic nose technology in early lung cancer diagnosis.Offers a reliable alternative to conventional diagnostic tools by combining intelligent algorithms with multidimensional sensor data.

**Abstract:**

In this study, a hybrid sensor-based electronic nose circuit was developed using eight metal-oxide semiconductors and 14 quartz crystal microbalance gas sensors. This study included 100 participants: 60 individuals diagnosed with lung cancer, 20 healthy nonsmokers, and 20 healthy smokers. A total of 338 experiments were performed using breath samples throughout this study. In the classification phase of the obtained data, in addition to traditional classification algorithms, such as decision trees, support vector machines, k-nearest neighbors, and random forests, the fuzzy logic method supported by the optimization algorithm was also used. While the data were classified using the fuzzy logic method, the parameters of the membership functions were optimized using a nature-inspired optimization algorithm. In addition, principal component analysis and linear discriminant analysis were used to determine the effects of dimension-reduction algorithms. As a result of all the operations performed, the highest classification accuracy of 94.58% was achieved using traditional classification algorithms, whereas the data were classified with 97.93% accuracy using the fuzzy logic method optimized with optimization algorithms inspired by nature.

## 1. Introduction

Motivation: Lung cancer (LC) has the highest mortality rate of all cancers worldwide. According to data from the World Health Organization (WHO), the global mortality rate of LC surpasses that of combined kidney, colorectal, and prostate cancers [1,2]. Although LC ranks second after bowel cancer in terms of the number of diagnosed cancers, it stands out as the form of cancer associated with the highest fatality rate. Globally, in 2020, there were 10 million fatalities attributed to cancer, and 19.3 million new cancer cases were reported, according to GLOBOCAN data. The two most common cancers in terms of incidence and mortality are LC and breast cancer, respectively. Figure 1 illustrates the incidence of new cancer cases and cancer-related fatalities in 2020, as reported by GLOBOCAN [3].

Studies have shown that while the five-year survival rate of individuals diagnosed with LC in the early stages can reach up to 90%, this rate drops below 10% in individuals whose disease is detected in the advanced stages [4]. Changes in the 5-year survival rates according to the stage at which LC is detected are shown in Figure 2 [5]. Various medical methods have been employed to examine individuals in the high-risk category for the early identification of LC. These approaches typically encompass a combination of saliva cytology [6,7], analysis of circulating tumor biomarkers [8,9], examination of blood protein structure [10,11], chest tomography [12,13], nuclear magnetic resonance [14], chest radiography [15], and low-dose computed tomography [16], among others. Because these methods exhibit restricted diagnostic capabilities when employed for an extended period, ongoing research is aimed at enhancing their performance. The limited success of imaging techniques for early-stage LC detection and mortality reduction has prompted researchers to focus on breath analysis in this field.

Proposed approach: Studies in the field of breath analysis aim to identify and measure volatile organic molecules (VOCs), which may be disease markers, in the breath.

Physiological or pathological changes in organ function can alter the composition of VOCs detected in exhaled breath [17]. When studies on breath analysis were examined, it was determined that in cases of febrile illness and sepsis, the pentane concentration in the breath was higher than that in healthy individuals [18]. Individuals with diabetes, a widespread global ailment, exhibit elevated levels of acetone and methyl nitrate in their exhaled breath [19,20]. Ammonia and di/trimethylamine are nitrogen-containing compounds present in the breath, and these volatile organic compounds are notably detected in the breath of individuals with kidney disease [21,22]. In addition, many studies have examined the VOCs in the breath associated with LC. In a study examining LC, which is the subject of this study, Peng et al. examined VOCs found in the breath of patients with cancer using gas chromatography–mass spectrometry (GC-MS) performed on four different types of cancer: LC, bowel cancer, breast cancer, and prostate cancer. As a result of the study, approximately 80% of the VOCs detected in the breath samples of healthy individuals and patients were found to be non-overlapping. Specifically, 34 VOCs were associated with colorectal cancer, 33 with lung cancer, 36 with prostate cancer, and 54 with breast cancer [23]. Poli et al. examined the aldehyde levels in the exhaled breath of individuals with LC. In their studies, they determined that propanal, butanal, pentanal, hexanal, heptanal, octanal, and nonanal aldehyde levels were different in the breath samples of the collection groups [24]. Jia et al. examined 25 studies in the literature in their review article on the breath content of individuals with LC. Biomarkers identified in a minimum of four studies were filtered and prioritized based on their frequency of occurrence. These VOCs were identified as toluene, propanol, isoprene, pentanol, acetone, hexane, ethyl benzene, 2-butanol, benzene, heptane, propanal, pentanal, ethanol, butanol, and benzene [25].

GC-MS is mostly utilized in these studies to ascertain breath content. In addition to these methods, ion flow tube mass spectrometry, laser absorption spectrometry, and infrared spectroscopy have been extensively utilized to identify VOCs in breath. However, these devices require methods that are expensive, and the tests are performed for a long time. Moreover, the preconcentration of breath samples is essential for enhancing the efficiency of VOC detection using these methods.

Hardware and data analysis approaches: compared with the above methods, e-noses are cheaper and easier to install and use. In addition, no concentration process was required before the experiments. Target gases can be detected and distinguished with high accuracy using the appropriate sensors [26,27]. The e-nose is an instrument inspired by the biological olfactory system of living organisms. Owing to its structure, which has both a chemical detection system and a data processing system, it can detect many simple and complex odors. These devices primarily comprise three components: a sensing unit, an electronic module, and a pattern-recognition component. An illustrated comparison of the e-nose and mammalian olfactory systems is shown in Figure 3 [28]. Numerous studies have utilized e-nose technology for breath analysis to detect lung cancer. To give examples of studies using MOS sensors, Binson et al. attempted to distinguish patients with lung cancer (LC) from other patients using an e-nose with metal oxide semiconductor (MOS) sensors, and reported sensitivity, specificity, and accuracy values of 91.3%, 84.4%, and 94.4%, respectively [29]. Morzarati et al. conducted their study using an e-nose consisting of a MOS sensor array, and 16 people, including lung cancer (*n* = 6) and control (*n* = 10). The researchers classified the data obtained from the e-nose in their study with 85.7% sensitivity, 100% specificity, and 93.8% accuracy [30]. Kort et al. used Aeonose brand e-nose device in their study with LC (*n* = 239) and control (*n* = 253) groups. The researchers classified the data with 91.3% sensitivity and 84.4% specificity rates as a result of data classification [31]. To give examples of pioneering studies using QCM sensors, Di Natale et al. used an e-nose equipped with quartz microbalance gas sensors coated with eight different metalloporphyrins to analyze 62 breath samples taken from 60 individuals, including lung cancer patients (*n* = 35), healthy controls (*n* = 18), and post-operative patients (*n* = 9). The researchers classified patients with cancer with 100% accuracy, healthy individuals with 94% accuracy, and post-operative patients with a 44% correct classification rate [32]. Gasparri and colleagues conducted a clinical study with 146 participants, including lung cancer patients (*n* = 70) and healthy controls (*n* = 76), using a gas sensor array formed by eight quartz microbalance sensors coated with metalloporphyrin. Using multivariate analyses, the researchers were able to distinguish cancer patients from healthy individuals with 81% sensitivity and 91% specificity. Additionally, a sensitivity of 92% has been reported for stage I lung cancer [33]. D’Amico and colleagues conducted their study with lung cancer patients (*n* = 28), healthy controls (*n* = 36), and patients with other lung diseases (*n* = 28). The classification achieved 79.3% success in distinguishing pathological conditions from healthy controls and 85.7% success between the two diseases. Sensitivity values exceeding 80% were reported for lung cancer detection: 85% compared to controls, 92.8% compared to other lung diseases, and 89.3% in triple classification [34]. Apart from these example studies, there are many studies in the literature where e-noses are used in the medical area [35,36,37,38]. A review of studies in the literature on lung cancer detection using electronic noses revealed notable differences in diagnostic performance depending on the type of sensor employed. In 11 studies where only metal oxide semiconductor (MOS) sensors were used, the mean sensitivity, specificity, and accuracy were 89% (range: 78–95%), 80% (range: 33–100%), and 87% (range: 73–97%), respectively. In contrast, in eight studies that utilized only frequency-based sensors, the mean sensitivity, specificity, and accuracy were 85% (range: 76–98%), 89% (range: 75–100%), and 89% (range: 75–100%), respectively. These findings suggest that while MOS-based systems tend to achieve slightly higher sensitivity, frequency-based systems may offer superior specificity, highlighting the importance of sensor selection in the design of electronic nose systems for clinical diagnostics [39,40,41]. In the literature, almost all e-noses used in studies on LC detection use a single type of gas sensor. This includes commercial e-noses and those developed by researchers. In this study, unlike other studies in the literature, two different types of sensors, MOS and Quartz Crystal Microbalance (QCM), were used to obtain both conductivity and frequency information. Although various studies have applied fuzzy logic-based approaches for data classification, these applications predominantly utilize fuzzy logic (FL) in conjunction with other computational techniques such as neuro-fuzzy networks, fuzzy k-nearest neighbors, or hybrid machine learning frameworks. In contrast, the approach adopted in this study relies solely on the fundamental components of fuzzy logic membership functions, rule bases, and inference mechanisms for data classification.

## 2. Materials and Methods

This section provides a detailed explanation of the developed e-nose system. Subsequently, the data preprocessing, feature extraction, and pattern recognition processes are described in detail in the following sections.

### 2.1. Experimental Setup

A block diagram of the e-nose circuit, which was used to perform breath analysis in this study, is shown in Figure 4. The experimental setup consisted of five components: dry air tubes, Teflon pipes, a vacuum pump (POM-VAK, Istanbul, Turkey), solenoid valves (JEL-PC, Ningbo, Zhejiang, China), a quartz crystal microbalance (QCM) sensor cell (TUBITAK MRC, Kocaeli, Turkey), an MOS sensor cell (Altinkaya, Ankara, Turkey), a sensor interface circuit, a control card, an analog digital data acquisition card USB-6218 Multifunction I/O Device (National Instruments, Austin, TX, USA), and a computer (Monster Tulpar T7, Istanbul, Turkey). To determine the MOS sensors (Figaro Engineering Inc., Osaka, Japan) to be used, studies in the literature were reviewed and analyzed. VOCs present at varying concentrations in the breath profiles of patients compared to those of healthy individuals were identified, and sensors were selected to detect these VOCs. The basic measuring circuit of the MOS sensors is shown in Figure 5. In this figure, $V_{C}$ and $V_{H}$ represent the circuit and heater voltages, respectively. The circuit and heater voltages of the MOS sensors used in this study are listed in Table 1. $R_{L}$ and $V_{RL}$ represent the load resistance connected in series with the sensor and the voltage value obtained across this resistance, respectively. The values of the load resistors for the MOS sensors are given in Table 2. Also, the detailed specifications for the MOS sensors are provided in Table 3. The data from the MOS sensors were transferred to the computer using a USB-6218 Multifunction I/O Device.

Quartz crystal microbalance (QCM) sensors are another type of gas sensor used in this study. In these sensors, a quartz disk is placed between two parallel gold electrode. Quartz, a piezoelectric material, undergoes mechanical deformation when an alternating current is applied to its electrodes. If the frequency of the applied current matches the resonance frequency of the QCM sensor, a standing wave associated with the mass of the sensor is generated in the resonator. It should be emphasized that QCM is not fundamentally a chemical sensor but rather an extremely sensitive mass transducer. However, after the sensor surface is coated with a chemically sensitive thin film that can selectively interact with the target analytes, it is transformed into a chemical sensor. When target gas molecules are absorbed by this functional layer, the resonance frequency of the sensor changes accordingly. Under certain conditions known as the small load approximation, the change in frequency can be approximately calculated and, as given in Equation (1), can be related to the mass change using the well-known Sauerbrey equation.(1)$$\;\Delta_{f}=-C_{f}\;\Delta_{m}$$

In this equation $\;\Delta_{f}$ represents the change in frequency (Hz), $\;C_{f}$ denotes the sensitivity factor of the employed quartz crystal (for instance, 56.6 Hz cm^2^/μg for a 5 MHz AT-cut quartz crystal at room temperature), and $\;\Delta_{m}$ signifies the change in mass per unit area. The sensitivity factor can be calculated as shown in Equation (2).(2) Cf=2nf02 μq pq12

In this equation, $\;\mu_{q}$ represents the shear modulus of quartz (2.947 × $10^{11}$ g/cm$s^{2}$), $\;p_{q}$ denotes the quartz density (2.648 g/${\mathrm{cm}}^{3}$), n is the harmonic number at which the crystal is driven, and $\;f_{0}$ is the fundamental resonant frequency of the quartz crystal. The dissipation factor is determined by taking the reciprocal of the resonance quality factor, as expressed in Equation (3).(3)$$D=\frac{1}{Q}=\frac{w}{\;f_{0}}$$

The equation, where *w* represents the bandwidth, measures the system’s damping. It can also be calculated as presented in Equation (4):(4)$$D=\frac{1}{\pi \;\Delta_{f}\tau }$$
where $\;\Delta_{f}$ represents the frequency change in Hertz, and *τ* denotes the decay constant of the quartz resonator. This can be interpreted as the ratio of the energy lost during each oscillation to the product of a constant and the total energy stored within the system, essentially comparing dissipated energy to conserved energy, as given in Equation (5).(5)$$D=\frac{\;E_{dissipated}}{2\pi E_{stored}}$$
Δ*f* is linked to the quantity of the sample that is adsorbed or desorbed, while *D* is associated with the sample’s stiffness and viscoelastic properties. The frequency change is measured and relayed to a computer via an appropriate electronic circuit. By analyzing this frequency shift, valuable insights about the target gases can be obtained. The basic working principle of a quartz crystal microbalance sensor can be seen in Figure 6.

QCM sensors were produced at TÜBİTAK Marmara Research Center. The researchers used AT-cut quartz crystals supplied by Klove Electronics (Westerlo, Belgium) in this study. Before application, a flow of dry air was used to remove the residual solvent from the QCM sensors. To ensure chemical selectivity, the crystal surfaces were coated with chemically sensitive films. In this study, vic-dioximes, poly(3-methylthiophene) (PMeT), axially substituted titanyl phthalocyanines (TiOPcs), vanadium pentoxide (V_2_O_5_), polypyrrole (PPy), and the copolymer of poly(3-methylthiophene) with polypyrrole (P3MT-co-PPy) were used as sensing materials. Owing to their high surface activities, chemical functionalities, and affinities for various cancer-related biomarkers, these coatings have been successfully applied in the literature as QCM sensing layers for the detection of volatile organic compounds (VOCs) [42,43,44,45]. The frequency values produced by the QCM sensors were detected using an internal frequency counter circuit and transferred to a computer. Thus, frequency information corresponding to the breath samples was acquired. Images of the QCM sensor chambers used in this study and a block diagram of the internal frequency counter circuit are shown in Figure 7 and Figure 8, respectively. Dry airflow was used to remove gas samples. The MOS and QCM sensors used in this study are shown in Figure 9. Figure 9a–d show the different types of MOS sensors used, while Figure 9e shows the type of QCM sensor used. Before and after all experiments, the entire system was cleaned with dry air containing 21% oxygen and 79% nitrogen. The flow rate of the dry air in the system was fixed at 10 L/min. Teflon pipes were used to accurately direct both dry air and breath samples to the appropriate sections. Teflon pipes do not react with most chemicals and do not retain odor molecules, thereby preventing any residual unwanted odor molecules from previous experiments from remaining on them. A vacuum pump powered by a 12 V DC supply was used to deliver breath samples to the sensors. The constant voltage of the vacuum pump ensured uniform delivery of the breath samples into the sensor chambers. The voltage supplied to the vacuum pump was determined during the installation phase of the e-nose system. During this phase, the volumes of the Tedlar bags and sensor chambers were collected, and the breath samples were introduced to the sensors at a flow speed of 3.6 L/min. Six solenoid valves were used in the experiments to send the breath samples and dry air directly. These valves, branded JELPC (JELPC, Zhejiang, Ningbo, China), had electrical working values of 24 V (DC) and 4.8 W. A control card circuit was designed to transfer the required energy to the valves, and a vacuum pump was used in our study. With the help of this control card, the valves and vacuum pump are activated for appropriate periods, and the experiments can be conducted as desired. A visual representation of the experimental setup is shown in Figure 10. The experiments were divided into four distinct phases. The first phase involved purging the system with dry air for 130 s. Following the initial cleaning, the vacuum pump transferred the collected breath to the QCM and MOS sensor chambers over a period of 40 s. Subsequently, all valves were closed, allowing a 30 s period for the sensors to react to the collected breath. Finally, the entire circuit underwent a 140 s cleaning cycle with dry air to prepare it for the next experiment. In this study, constant temperature and humidity levels were maintained by using air conditioning in the laboratory room where the experiments were conducted.

The durations of the phases were predetermined during the design of the experimental setup. In preliminary trials, longer durations were tested; however, once the sensor responses reached a stable trend, extending the phase time did not yield additional benefits. Therefore, the transition to the next experimental stage was initiated after this stabilization point. A randomly selected sample from the data of the MOS and QCM sensors is shown in Figure 11. In this Figure, the behavior of the sensor data at the stages of the experiment can be seen.

### 2.2. Collection of Breath Samples

This study involved 338 experiments utilizing breath samples obtained from 40 healthy volunteers and 60 lung cancer patients treated at the Hospital of the Faculty of Medicine, Karadeniz Technical University. Prior to the work, all required ethics committee documents were obtained from Karadeniz Technical University, and the ethics approval number assigned was 24237859-517. After LC diagnosis of each patient volunteer, participants in the study were confirmed according to bronchoscopy and biopsy results, and breath samples were collected. Information on all volunteers from whom breath samples were collected and the cancer stage along with lung cancer cell types of the study participants are presented in Table 4 and Figure 12, respectively.

The tumor stages of the lung cancer patients participating in this study were evaluated using the 7th edition of the TNM classification system, which assesses Tumor Extent (T), Regional Lymph Node Metastases (N), and Distant Metastasis (M) as defined by the American Joint Committee on Cancer (AJCC). Prior to breath sample collection, the patients did not undergo any surgical procedures. Exhaled breath samples from the participants were collected utilizing 5 L Tedlar sampling bags manufactured by CEL Scientific, Cerritos, CA, USA. Due to the gas-impermeable nature of Tedlar bags, the composition of the collected breath samples remains stable, allowing for accurate experimental analysis without alteration of their contents. In this study, Tedlar bags fitted with a polypropylene connector were used to control the direction of exhalation. A schematic representation of the breath sample collection is shown in Figure 13. Breath samples were collected by exhaling through the inlet marked as number 1. If the breath was to be directed through the inlet marked as number 2 instead, the pipeline labeled number 3 needed to be closed. When this pipeline remained open, exhaled air entering through inlet number 1 was redirected outward through pathway number 3, preventing it from entering the bag. To ensure that the sampled air originated from the alveolar region of the lungs rather than from the upper respiratory tract, a timing-based approach was utilized. A deep exhalation was considered to last approximately six seconds. Based on this estimation, the pipeline labeled number 3 was kept open during the first three seconds of exhalation, allowing the initial portion of the breath—representing air from the upper respiratory tract—to be released. After three seconds, the pipeline was closed, enabling the remaining air—originating from the alveolar region—to be collected in the Tedlar bag for analysis. To participate in the experiments, individuals were instructed to abstain from food intake overnight and to provide their breath samples while in a fasted state. Additionally, they were required to avoid smoking for at least two hours prior to sample collection. All breath samples were collected between 8 and 11 a.m. to ensure stability from the beginning of the study. The healthy volunteers who participated in this study were not pregnant and did not have any other lung disease. Half of the healthy volunteers were healthy smokers. All volunteers participated in this study voluntarily.

### 2.3. Analysis of Data and Extraction of Features

Before feature extraction, the data were preprocessed. To ensure accurate visualization of the obtained data and obtain beneficial information from these data, reference correction was applied to the data obtained from both the MOS and QCM sensors using Equations (6) and (7), respectively.(6)$$\;v_{n,s,r}\left(t\right)=v_{n,s}\left(t\right)-v_{n,s}\left(130\right)$$(7)$$f_{n,s,r}\left(t\right)=f_{n,s}\left(t\right)-f_{n,s}\left(130\right)$$

In these equations, $v_{n,s}\left(t\right)$ represents the raw sensor data, $v_{n,s}\left(130\right)$ represents the sensor data obtained at the first moment that the collected sample was delivered to the sensors, and $\;v_{n,s,r}\left(t\right)$ represents the data for which the value of $v_{n,s}\left(130\right)$ is set to zero. Representative examples of the raw MOS sensor data and the corresponding reference-corrected data are presented in Figure 14. $f_{n,s}\left(t\right)$ represents the raw frequency data, $f_{n,s}\left(130\right)$ represents the resonance frequency value of the sensors, and $f_{n,s,r}\left(t\right)$ denotes the data for which the value of $f_{n,s}\left(130\right)$ is set to zero. Representative examples of the raw QCM sensor data and the corresponding reference-corrected data are presented in Figure 15. Upon reviewing the e-nose studies presented in the literature, it was determined that the conductivity information of the MOS sensor was used rather than the voltage information obtained from the load resistance connected in series to the MOS sensors. In this study, the voltage information derived from the load resistances connected to the sensors was converted into conductivity data using Equation (8).
Figure 14(**a**) Raw data; (**b**) reference-corrected data of MOS sensors.
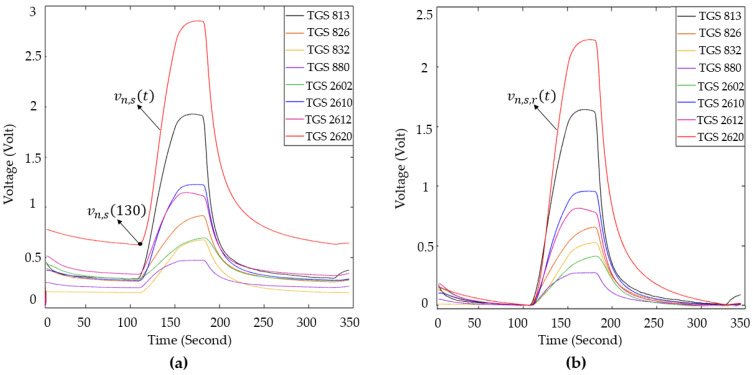

Figure 15(**a**) Raw data; (**b**) reference-corrected data of QCM sensors.
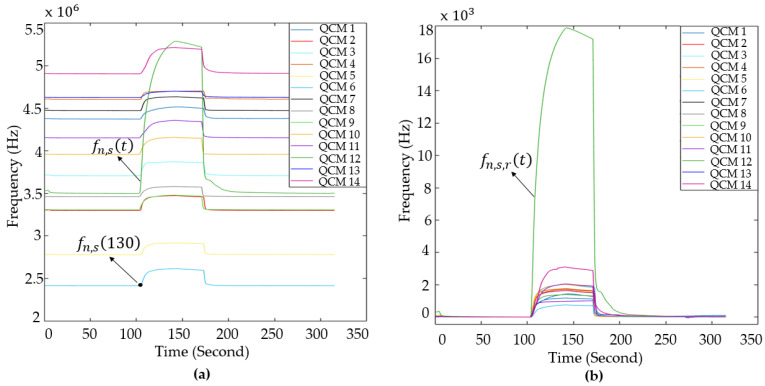

(8)$$G_{n,s,r}\left(t\right)=\frac{V_{n,s,r}\left(t\right)}{\left(V_{c}-V_{n,s,r}\left(t\right)\right)R_{L,s}}$$

In this equation, $V_{c}$ represents the supply voltage of the sensors, $R_{L,s}$ represents the resistance values connected in series to the MOS sensors, and $G_{n,s,r}\left(t\right)$ represents the conductivity information of the MOS sensors. Figure 16 displays example data from the reference-corrected MOS sensor data and the conductivity of the MOS sensor data. Matlab2022b software was used for all operations required for data processing in this study. As part of this study, 338 experiments were conducted using breath samples obtained from all participating volunteers. Of these experiments, 219 were conducted using breath samples from individuals diagnosed with LC, and 119 used samples from healthy volunteers. In this study, several features were extracted, including the maximum, variance, mean, kurtosis, skewness, and gradient of the data across specified temporal intervals, such as:The area between TGS826 and TGS832 sensor data in the interval of 130–170 sThe area between TGS813 and TGS2620 sensor data in the interval of 130–170 sThe area between TGS880 and TGS2610 sensor data in the interval of 130–170 sThe slope of all sensor data in the interval of 130–145 sThe slope of all sensor data in the interval of 145–170 sThe slope of sensor data in the interval of 170–200 sThe slope of all sensor data in the interval of 200–215 sThe slope of sensor data in the interval of 200–230 sThe area under all sensor data in the interval of 145–215 s

**Figure 16 sensors-25-05271-f016:**
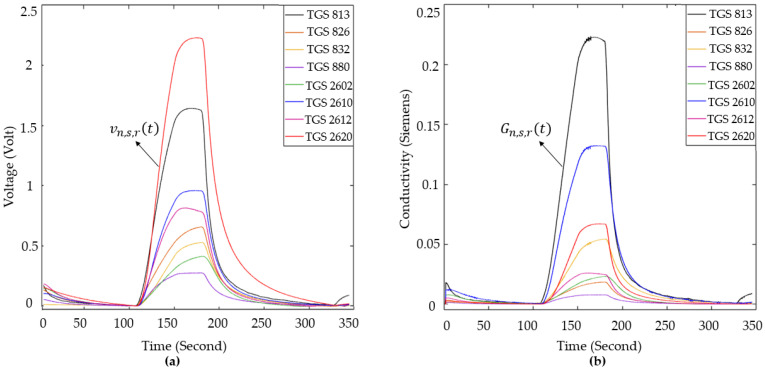
(**a**) Raw data; (**b**) reference-corrected conductivity data of MOS sensors.

As a result of using the specified features, the dimension of the feature matrix using the data of the MOS sensors was (338 × 91). Here, 338 denotes the total number of experiments conducted, while 91 corresponds to the total number of extracted features. For instance, considering that there are eight MOS sensors, calculating the maximum value for each sensor results in eight individual feature values, which collectively contribute eight columns to the feature matrix. By combining the contributions from all feature types employed in this study, the final feature matrix size was obtained as (338 × 91).

As a result of the examination of QCM sensor data, it was decided to use the following attributes.

The area between the QCM6 and QCM14 sensor dataThe area between the QCM2 and QCM11 sensor dataThe area between the QCM9 and QCM11 sensor dataThe area between the QCM14 and QCM7 sensor dataThe slope of the QCM11 sensor data between 100 and 140 sThe slope of the QCM6 sensor data between 100 and 140 sThe maximum value of the QCM5 sensor was used.

The size of the feature matrix created using the QCM sensors was (338 × 7). The feature matrices obtained in this study were classified without reducing the size of the feature matrix and by reducing the size of these matrices using linear discriminant analysis (LDA) and principal component analysis (PCA) algorithms. When using these algorithms, care was taken to ensure that the data with a reduced size contained at least 90% of the information contained in the main data. The initial dimensions of the feature matrix derived from the MOS sensor data were (338 × 91), which were reduced to (338 × 11) and (338 × 2) through the application of LDA and PCA, respectively. The initial dimensions of the feature matrix derived from the QCM sensor data were (338 × 7), and these dimensions were reduced to (338 × 4) and (338 × 3) by applying the LDA and PCA methods, respectively. Following these processes, a feature matrix with dimensions of (338 × 98) was generated by integrating the original feature matrices derived from the MOS and QCM sensors. In the remainder of this paper, this feature matrix is referred to as the hybrid feature matrix. The dimensions of the hybrid feature matrix were reduced using LDA and PCA. The size of the hybrid feature matrix was reduced to (338 × 14) using the PCA method and to (338 × 3) using the LDA method. The scatter graphs created using the first two features of the dimension-reduced hybrid feature matrix and the percentage contribution of eigenvalues for PCA and LDA are shown in Figure 17 and Figure 18, respectively.

## 3. Data Classification and Results

This section presents the data classification procedures employed in this study. In the first stage, breath samples were classified using conventional machine learning methods, including decision trees (DT), support vector machines (SVM), k-nearest neighbors (k-NN), and random forests. In the second stage, the classification was performed using a fuzzy logic approach, which was optimized with various optimization algorithms, based on a feature matrix whose dimensionality had been reduced via linear discriminant analysis (LDA).

### 3.1. Classification with Traditional Methods

In this study, the data of LC, healthy non-smokers (HnS), and healthy smoker (HS) volunteers were classified into three classes using classification algorithms. The created feature matrices were classified using decision tree (DT), random forest (RF), k-nearest neighbor (k-NN), support vector machine (SVM), and fuzzy logic (FL) algorithms, without reducing their dimensions and by reducing their dimensions using PCA and LDA methods. The classification results are presented in Table 5, and the feature matrices in question are labeled using abbreviations that indicate the sensor data features utilized and the dimension reduction algorithm applied to reduce the dimensions of these matrices. For instance, the feature matrix derived from MOS sensor data without dimension reduction is denoted as MOS (in the rest of this study, the names of the feature matrices are underlined), the feature matrix obtained by reducing its dimension using the PCA method is denoted as PCA(MOS), and the feature matrix obtained by reducing its dimension using the LDA method is denoted as LDA(MOS). A 5-fold cross-validation method was used to calculate the performance of the classifiers used in this study. The classification process was repeated 10 times for each classification algorithm for each feature matrix, and the results of each classification process were provided, including the average accuracy, highest accuracy, lowest accuracy, and standard deviation values. In addition to the results presented in Table 5, confusion matrices were created for the classification of feature matrices whose dimensions were not reduced. In Table 5 bold values indicate the highest classification accuracy obtained for each feature. The confusion matrices related to the classification results of the MOS, QCM, and MOS + QCM feature matrices are shown in Figure 19. When these confusion matrices are examined, the effect of the sensor type used on the classification success is observed more clearly. As shown in Figure 19, the feature matrix derived using MOS sensors provided a higher classification accuracy than the feature matrix obtained from QCM sensors. One of the main motivations of this study was to increase the classification success by combining different types of sensor data. It is clearly seen in Figure 19 that this process is successful. It is clearly seen in Figure 19 that the accuracy obtained by classifying the MOS + QCM feature matrix, which is the feature matrix created by combining the MOS and QCM features, is higher than the classification success obtained by using the MOS and QCM features separately. In addition, to determine the impact of the PCA and LDA algorithms on classification success, confusion matrices of dimension-reduced data were calculated. The confusion matrices shown in Figure 20 and Figure 21 show the classification results for the feature matrices whose dimensions were reduced using the PCA and LDA algorithms, respectively. Upon analyzing these results, it can be seen that the feature matrix derived by combining the MOS and QCM features provides the best classification accuracy, even when its dimensions are reduced. When Figure 14 and Figure 15 are examined, it is observed that the LDA algorithm has a greater positive effect on classification success than the PCA algorithm.

### 3.2. Classification with Fuzzy Logic Method

Another algorithm used for data classification in this study was the FL method. The distribution graph formed as a result of reducing the dimensions of the MOS + QCM with LDA in Figure 17b reveals that this feature matrix can be classified using the FL method. In the FL method, the more information we have about the problem to be solved, the more accurate the membership functions to be used and the rule table to be created are. Therefore, the distribution graph was carefully examined before the classification. It was decided to use two membership functions for the HnS and HS classes, while it was decided to use three membership functions for the cancer patients class. The regions where the membership functions to be used were initially placed in the distribution graph are shown in Figure 22. This initial placement process was performed such that the distribution graph of each class was divided into equal parts.

One of the most important issues in applications where the FL method is used is the correct creation of a rule table. Determining the rule table appropriately is the most important factor that provides the correct solution to the problem being addressed. To determine the rule table, the system being examined must be examined in detail by experts, and all its details must be mastered. In this study, the rule table was constructed by analyzing the intersection sets of the membership functions. For each intersection, the rule was assigned to the class that had the largest number of data points within that region. Because seven membership functions are used for both LD1 and LD2 features, the size of the rule table created is (7 × 7), and the number of rules is 49. The rule table created for the classification using FL is presented in Table 6. In this table, “O” represents healthy smokers, while “+” and “◊” represent healthy non-smokers and lung cancer patients, respectively. For instance, if data from the LD1 feature belongs to the “$O_{2}$” fuzzy set and the data from the LD2 feature belongs to the “${+}_{1}$” fuzzy set, the breath of the individual belongs to the “+” class.

Another important process that increased the success of a study and applications performed using the FL method was the appropriate selection of the parameters of the membership functions used [46]. The parameters of the membership functions determine their positions and shapes, and these have been chosen appropriately to enhance system performance [47,48]. The information on the membership functions used in this study is listed in Table 7.

The parameters of the membership functions used in this study were determined using five different nature-inspired optimization algorithms. In this process, the dataset to be classified was divided into five parts. The first four parts were used as training data. The parameters of the membership functions were initially plotted on a scatter plot using the training data. These parameters were optimized using five different nature-inspired optimization algorithms. The results obtained using the optimized membership functions were compared with the real class values. If this comparison result does not converge sufficiently, the optimization algorithm updates the parameters of the membership functions. This update continues until the results obtained using the test data are as close as possible to the actual results. For each of the four training data and one test data combinations, this process was repeated 10 times, and the results were recorded. The process described above is repeated such that the second dataset is the test data. This process continued until each piece of the dataset was divided into five pieces and used as the test data. The performance of the classification process was recorded by calculating the maximum, minimum, mean, and standard deviation of the classification accuracy obtained as a result of this process. A visual representation of the implemented method is shown in Figure 23. In this study, the parameters of the membership functions were determined using nature-inspired optimization algorithms. Nature-inspired optimization algorithms are based on the remarkable and complex behaviors observed in natural systems. In recent years, the growing complexity of optimization problems has motivated researchers to investigate efficient algorithms that emphasize decentralized and self-organizing systems for solving problems [49,50]. Metaheuristic algorithms are inspired by physical phenomena, biological evolution, and the behavior of organisms such as fish, termites, birds, and ants. In this study, the genetic algorithm (GA), particle swarm optimization (PSO), simulated annealing (SA), invasive weed optimization (IWO) algorithm, and artificial ecosystem-based optimization (AEO) algorithms were used to optimize the membership functions [51,52,53]. Information regarding these algorithms is provided in Table 8.

In light of these operations, the details of the classification results obtained with FL through the optimization of the membership functions and the confusion matrices corresponding to the top three classification accuracies are presented in Table 9 and Figure 24, respectively. The classification process was repeated 10 times with each optimization algorithm and membership function, and the results of each classification are given as the mean accuracy, highest accuracy, lowest accuracy, and standard deviation value. The hyperparameters of the optimization algorithms used in this study can be seen in Table 10. When the results in Table 9 are examined, it is observed that the highest classification success is achieved by using the GA-generalized bell-shaped pair using the FL classifier. As a result of this process, in which the highest classification accuracy was achieved, the final states of the generalized bell-shaped membership functions created by the optimization algorithm on the distribution graph are shown in Figure 25.

When the results listed in Table 9 are examined, the best classification result of all the optimization algorithms used was achieved using the generalized bell-shaped membership function. In this study, in addition to the membership function and optimization algorithm that provides the best classification process, the performances of all the optimization algorithms used and all membership functions were examined. The performances of each optimization algorithm using Gaussian, generalized bell-shaped, pi-shaped, trapezoidal, and triangular membership functions are shown in Figure 26,Figure 27,Figure 28,Figure 29 and Figure 30, respectively.

## 4. Discussion

The results of this study highlight the advantages of integrating different sensor types into electronic nose (e-nose) systems for lung cancer detection. Most previous research in the field, whether using commercial or custom-made sensors, has typically focused on e-noses that employ gas sensors of a single category. Although such systems have shown promising results, their discriminatory power has generally been limited by the detection range and selectivity inherent to the specific sensor technology. The current study overcomes this limitation by introducing a hybrid e-nose configuration that combines metal oxide semiconductor (MOS) and quartz crystal microbalance (QCM) sensors, thereby expanding the detection spectrum and enhancing the overall detection capability. Comparative analyses of separate and combined feature matrices confirmed that sensor fusion led to a marked improvement in the classification performance, and the hybrid MOS+QCM feature matrix surpassed the individual forms. Furthermore, to place the current findings in a broader scientific context, a review and discussion of the relevant literature is necessary. In this regard, studies on hybrid e-nose systems and fuzzy logic-based classification approaches are addressed, with their similarities and differences evaluated. The second key finding addresses the impact of dimensionality-reduction techniques on classification accuracy. While traditional classifiers achieved competitive results on the original feature matrices, the application of Linear Discriminant Analysis (LDA) provided a notable performance boost, particularly for the hybrid dataset. This result demonstrates that LDA not only reduces redundancy in high-dimensional sensor data but also emphasizes features that maximize interclass separation. Interestingly, the data structure reflected by LDA inspired the use of Fuzzy Logic (FL) as a standalone classifier. Unlike many previous studies, fuzzy logic was not combined with neural networks or other machine learning models but was implemented in its basic form, consisting of membership functions, a rule base, and an inference engine, without additional computational frameworks. The optimization of membership functions using five different nature-inspired algorithms underscores the importance of parameter tuning in fuzzy systems. The high classification accuracy of 97.93% achieved with the optimized LDA(MOS+QCM) feature matrix demonstrates the synergistic effect of dimensionality reduction, sensor fusion, and optimized fuzzy inference. These results indicate that when carefully designed, fuzzy sets and rule bases can compete with—and in some cases surpass—more complex hybrid machine learning systems. This is particularly significant in scenarios where interpretability and computational efficiency are critical. From an application standpoint, these findings have practical implications for the development of portable and cost-effective diagnostic tools for early-stage lung cancer detection. The use of MOS and QCM sensors within a single compact e-nose platform could enable more reliable breath analysis in clinical and nonclinical settings. The developed e-nose is currently being used in an experimental setup, and work is ongoing to develop the smallest possible prototype. Additionally, in the ongoing development of the e-nose, a section is planned to be added to allow breath samples to be collected without using Tedlar bags. This modification makes the device easily portable. The only limiting factor is the need for a dry air cylinder; even if the e-nose circuitry is miniaturized and made portable, a dry air cylinder is still required at the experimental location. However, several limitations should be considered. Although the dataset was balanced in terms of classes, it was relatively small. Larger multicenter studies are needed to confirm the generalizability of these results. Although environmental factors, such as humidity, temperature, and background volatile organic compounds, were controlled in this study, robust strategies to compensate for these variables are necessary for real-world applications. Future research should investigate the integration of other sensor technologies, such as electrochemical or optical sensors, to further expand the detection range. Moreover, adaptive fuzzy systems that can update membership functions in real time based on incoming data streams may enhance long-term performance in dynamic environments. Finally, longitudinal studies that follow patients over time could reveal whether the proposed system can detect early-stage disease progression and responses to treatment. Overall, this study provides strong evidence that combining heterogeneous gas sensors with optimized fuzzy logic classification can significantly improve the accuracy of lung cancer detection, offering practical potential for both scientific innovation and real-world applications.

## 5. Conclusions

In the literature, almost all e-noses used in studies on LC detection use a single type of gas sensor. This includes commercial e-noses and those developed by researchers. In this study, a hybrid sensor-based e-nose circuit was developed using eight MOS sensors and 14 QCM sensors. Experiments were performed in an e-nose circuit with breath samples collected from 60 LC patients, 20 HnS, and 20 HS volunteers. The data obtained from the experiments were examined, and the features that best distinguished the data of the LC-HnS-HS volunteers from each other were determined. The feature matrices created in this study were initially classified using traditional classification methods, such as DT, k-NN, SVM, and RF. When the MOS and QCM feature matrices were classified separately, the highest classification accuracies obtained were 81.54% and 73.18%, respectively. After the separate classification of the MOS and QCM feature matrices, the hybrid feature matrix (MOS+QCM) created by combining these feature matrices was classified. The hybrid feature matrix was classified with a classification accuracy of 85.26%. The classification results support the idea at the beginning of this study that the combined use of different types of sensors could increase the detection of lung cancer. In the initial stage, the created feature matrices were classified without reducing their dimensions. Then, to determine the impact of PCA and LDA dimension reduction algorithms on the classification success, data classification was performed by reducing the dimension of the extracted features, and the LDA dimension reduction algorithm was observed to have a significant effect on the classification accuracy. Figure 31 presents a comparative illustration of the impact of different feature matrix types and dimensionality reduction techniques on classification accuracy. In addition, the distribution between classes obtained by reducing the size of the feature matrix using the LDA method gave rise to the idea that the FL method could also be used as a classifier. In the classification with FL, the distribution matrix shown in Figure 12b was carefully examined, and the most appropriate rule table was created. After creating the rule table, the membership functions, which are another basic element of the FL algorithm, were determined. Five different types of membership functions (Gaussian, generalized bell-shaped, pi-shaped, trapezoidal, and triangular) were used to classify data using the FL method. In FL classification, the positions of the membership functions were arranged in a way that would separate the data in the scatter plot at equal intervals. The parameters that formed the shapes of these membership functions were assigned randomly. Subsequently, these membership functions were optimized using five different nature-inspired optimization algorithms (GA, PSO, SA, IWO, and AEO). With this optimization process, both the positions and shapes of the membership functions were ideal. Consequently, the LDA(MOS+QCM) feature matrix was classified with a 97.93% classification accuracy. The effects of different fuzzy sets and optimization algorithms used to optimize the FL algorithm should not be overlooked. An analysis of Table 7 clearly reveals that the membership functions and optimization algorithms used affect the classification accuracy. Figure 32 presents a comparative illustration of the impact of different membership functions and nature-inspired optimization algorithms on classification accuracy. Although various studies have applied fuzzy logic-based approaches for data classification, these applications predominantly utilize fuzzy logic in conjunction with other computational techniques such as neuro-fuzzy networks, fuzzy k-nearest neighbors, or hybrid machine learning frameworks. In contrast, the approach adopted in this study relies solely on the fundamental components of fuzzy logic—namely, membership functions, rule base, and inference mechanism—for data classification.

When the results of this study are compared with those of pioneering QCM-based clinical studies, the advantages of the hybrid approach become more apparent. Di Natale et al. achieved 100% accuracy for cancer, 94% for healthy controls, and 44% for post-surgery patients using QCM sensors functionalized with metalloporphyrins [32]. Gasparri and colleagues reported 81% sensitivity and 91% specificity (92% for stage I lung cancer) in a cohort of 146 individuals [33]. D’Amico et al. demonstrated sensitivities above 80% in distinguishing patients with cancer, healthy controls, and individuals with other lung diseases [34]. In this context, the hybrid MOS+QCM system developed in this study, which achieved an accuracy of 97.93%, a sensitivity of 98.64%, and a specificity of 96.64%, represents a significant improvement over systems relying solely on QCM sensors and highlights the advantages of integrating different sensor technologies.

## Figures and Tables

**Figure 1 sensors-25-05271-f001:**
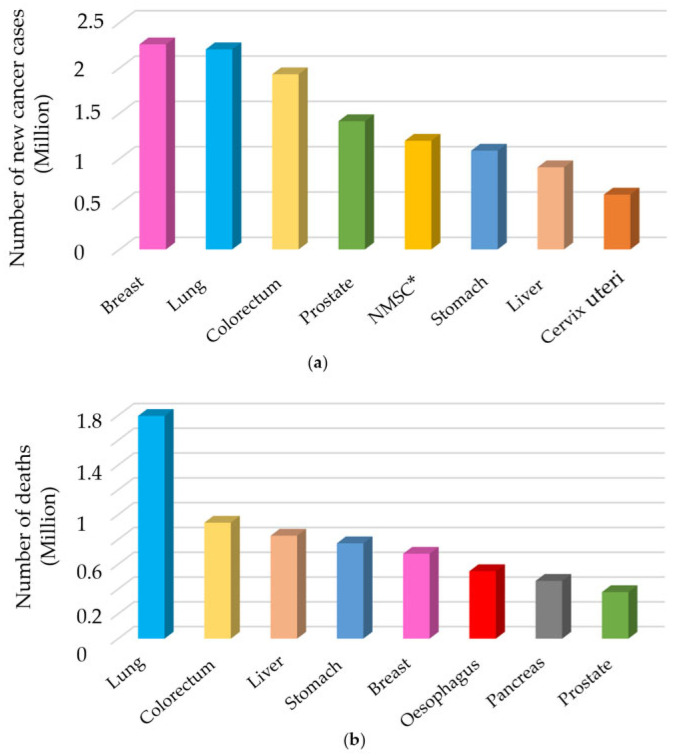
(**a**) The incidence of new cancer cases and (**b**) fatalities attributed to cancer in the year 2020. * NMSC: Non-melanoma skin cancer.

**Figure 2 sensors-25-05271-f002:**
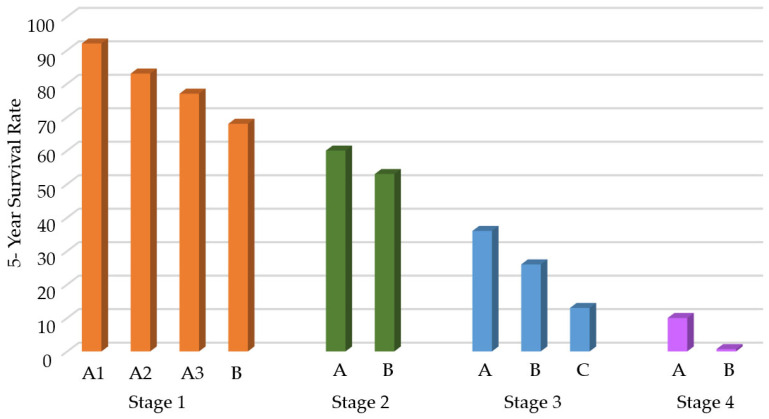
Changes in 5-year survival rates according to the stage at which lung cancer is detected.

**Figure 3 sensors-25-05271-f003:**
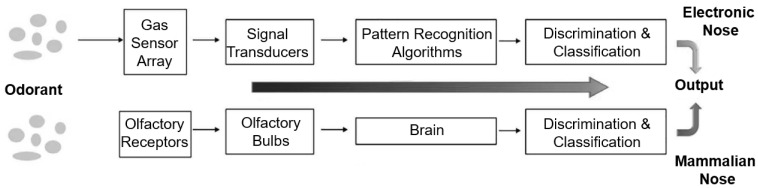
An illustrative diagram displaying the structure of the mammalian nose in comparison with an e-nose.

**Figure 4 sensors-25-05271-f004:**
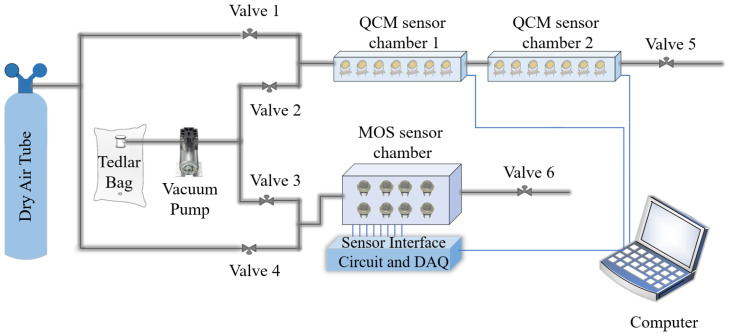
Schematic representation of the e-nose circuit.

**Figure 5 sensors-25-05271-f005:**
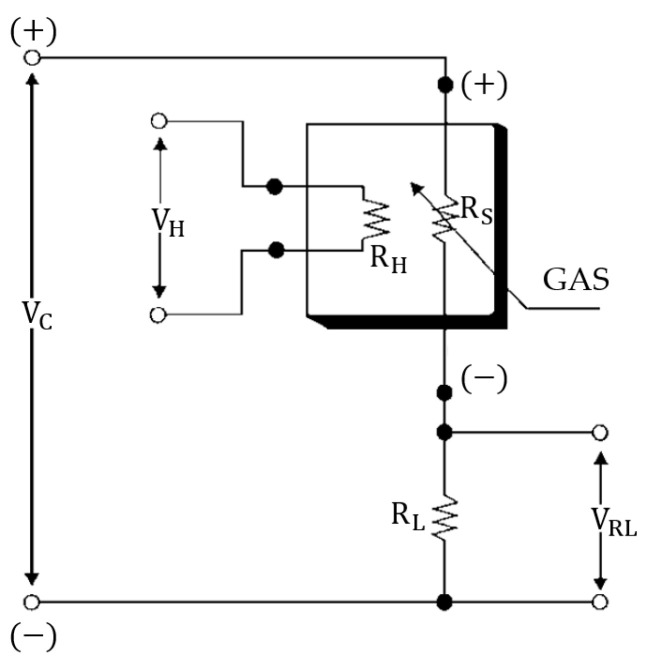
Basic measuring circuit of MOS sensors.

**Figure 6 sensors-25-05271-f006:**
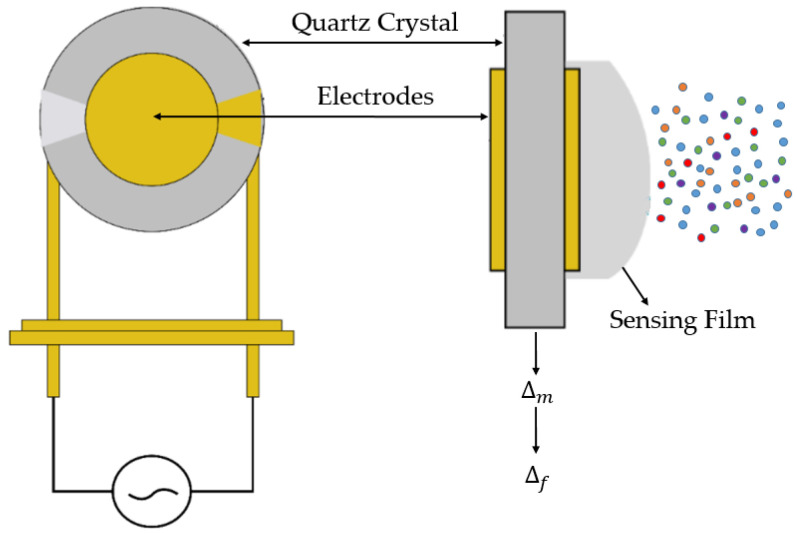
Basic working principle of quartz crystal microbalance sensor.

**Figure 7 sensors-25-05271-f007:**
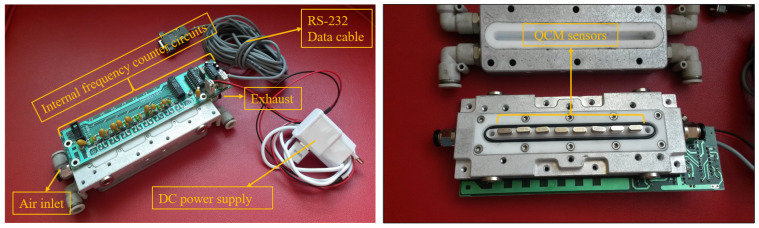
QCM sensor chamber.

**Figure 8 sensors-25-05271-f008:**
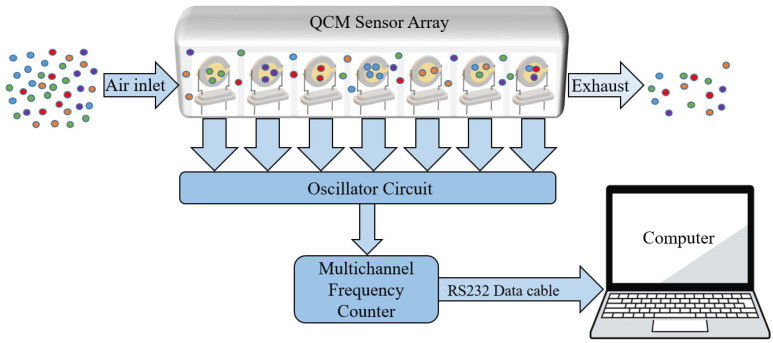
Block diagram of internal frequency counter and QCM sensor chamber.

**Figure 9 sensors-25-05271-f009:**
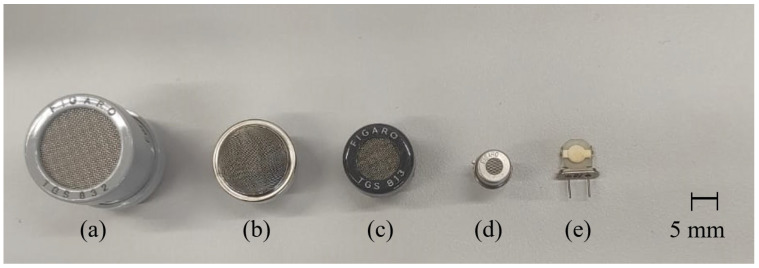
QCM and MOS sensors utilized in this study. (**a**–**d**) different types of MOS sensors used, (**e**) the type of QCM sensor used.

**Figure 10 sensors-25-05271-f010:**
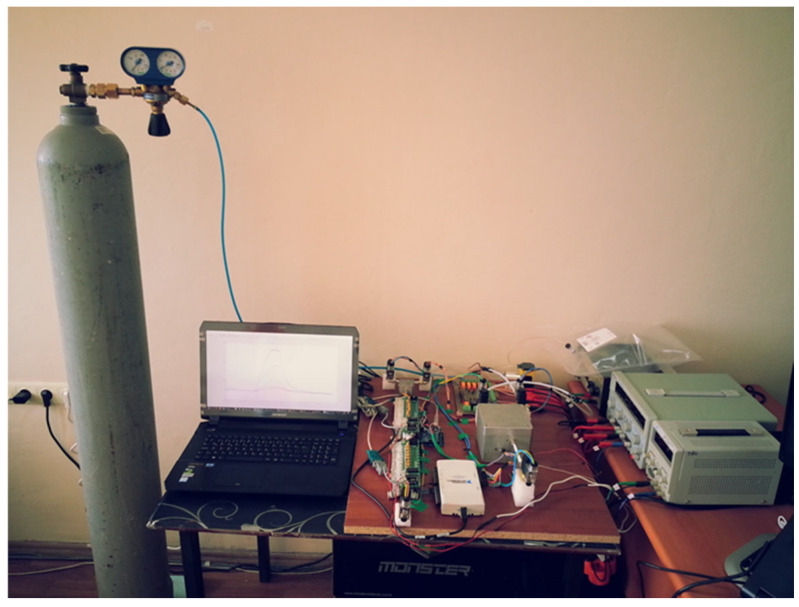
Developed e-nose circuit.

**Figure 11 sensors-25-05271-f011:**
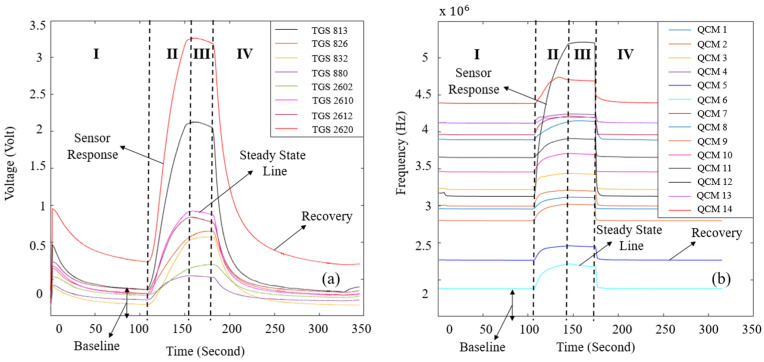
(**a**) The signals of the MOS sensor and (**b**) QCM sensor generated as a result of the experiment.

**Figure 12 sensors-25-05271-f012:**
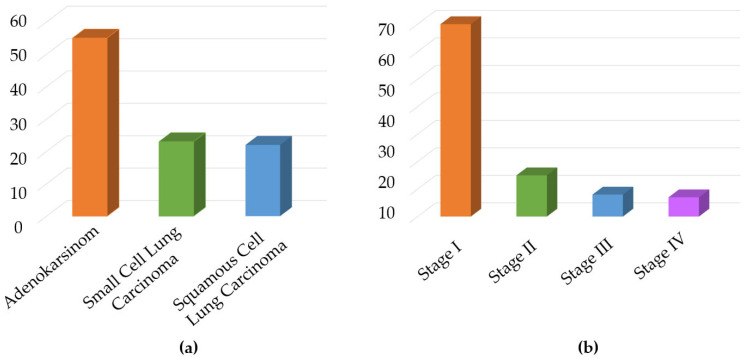
(**a**) Distribution of patients based on the histological type of cancer and (**b**) distribution of patients according to the cancer stage.

**Figure 13 sensors-25-05271-f013:**
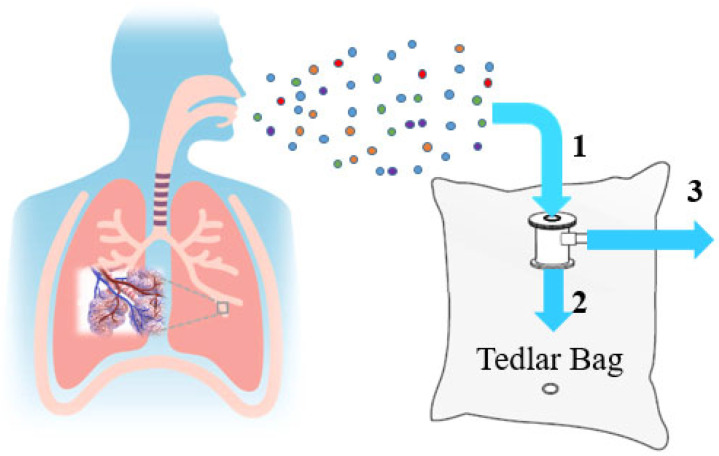
A schematic representation of the breath sample collection.

**Figure 17 sensors-25-05271-f017:**
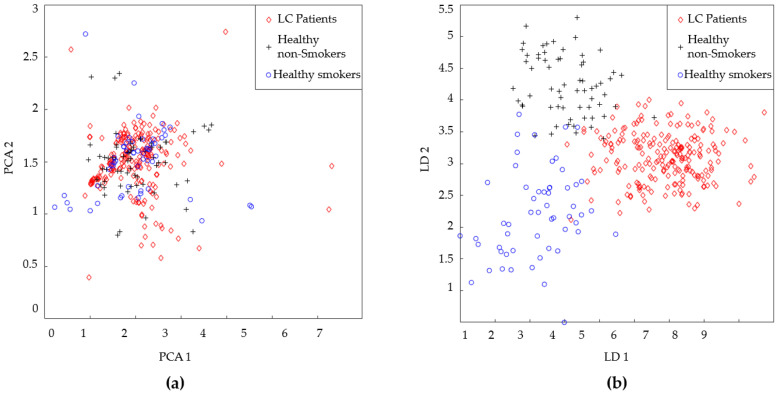
Scatter plot of the first two features of the dimension-reduced hybrid feature matrix, (**a**) the dimension of which is reduced by PCA; (**b**) the dimension of which is reduced by LDA.

**Figure 18 sensors-25-05271-f018:**
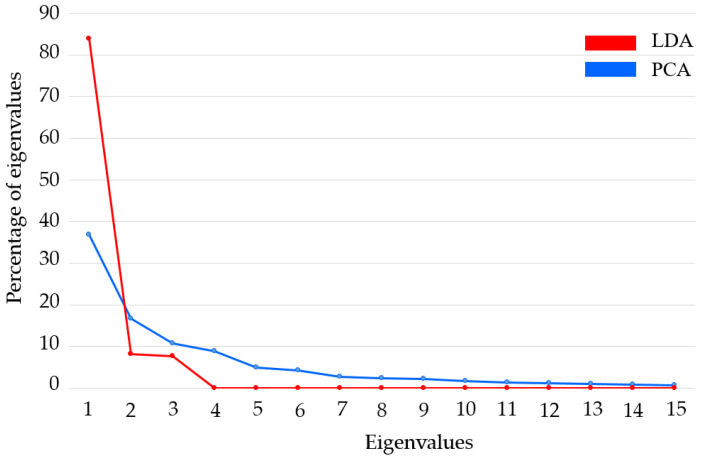
Percentage contribution of eigenvalues for PCA and LDA.

**Figure 19 sensors-25-05271-f019:**
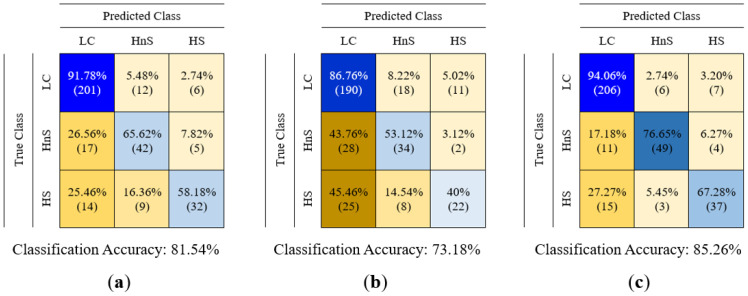
Confusion matrices obtained as a result of classification of (**a**) MOS, (**b**) QCM, (**c**) MOS + QCM feature matrices.

**Figure 20 sensors-25-05271-f020:**
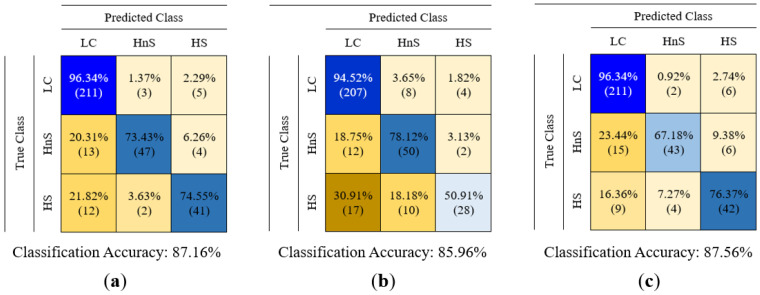
Confusion matrices obtained as a result of classification of (**a**) PCA(MOS), (**b**) PCA(QCM), (**c**) PCA(MOS + QCM) feature matrices.

**Figure 21 sensors-25-05271-f021:**
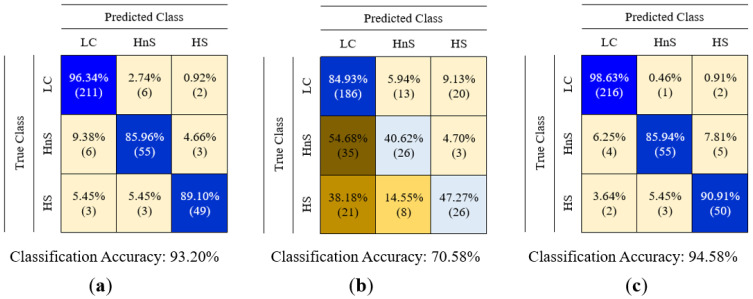
Confusion matrices obtained as a result of classification of (**a**) LDA(MOS), (**b**) LDA(QCM), (**c**) LDA(MOS + QCM) feature matrices.

**Figure 22 sensors-25-05271-f022:**
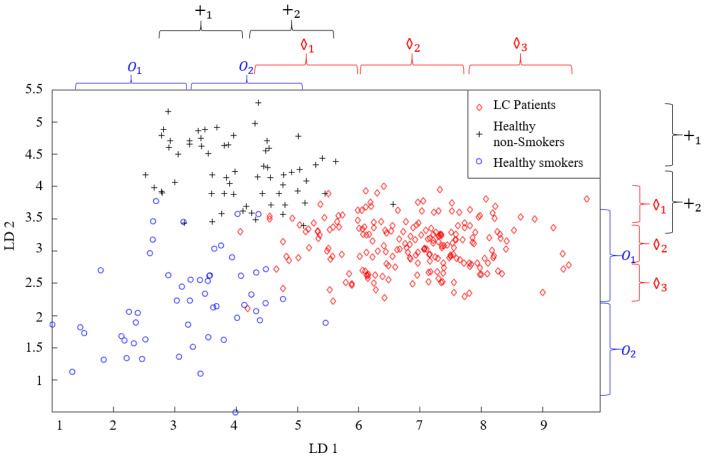
The initial placement of fuzzy sets on the scatter plot.

**Figure 23 sensors-25-05271-f023:**
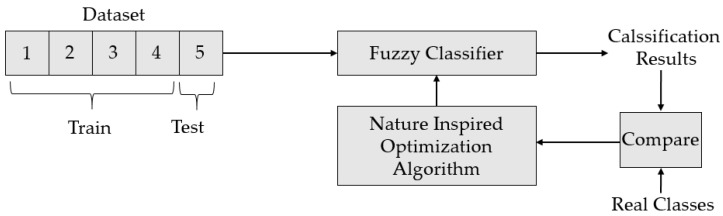
Optimizing the parameters of fuzzy sets.

**Figure 24 sensors-25-05271-f024:**
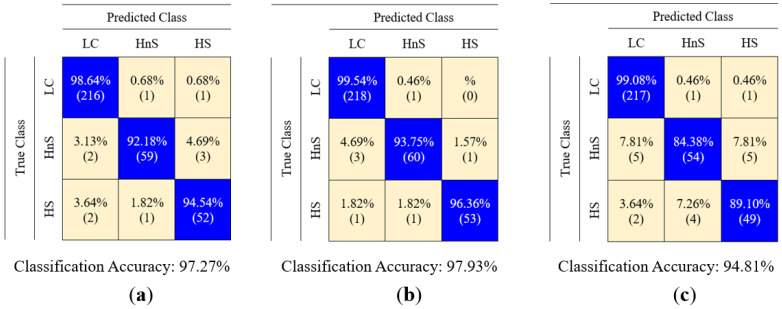
Confusion matrices corresponding to the top three classification accuracies: (**a**) PSO-Gaussian, (**b**) GA-generalized bell-shaped, (**c**) GA-triangular.

**Figure 25 sensors-25-05271-f025:**
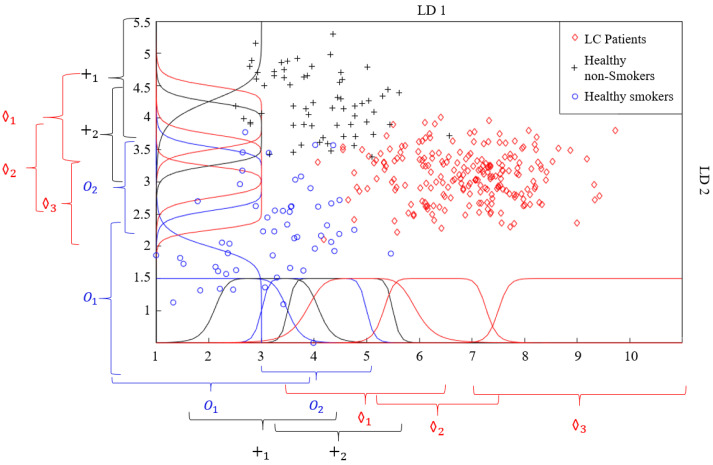
Placement of optimized generalized bell-shaped membership functions on the scatter plot.

**Figure 26 sensors-25-05271-f026:**
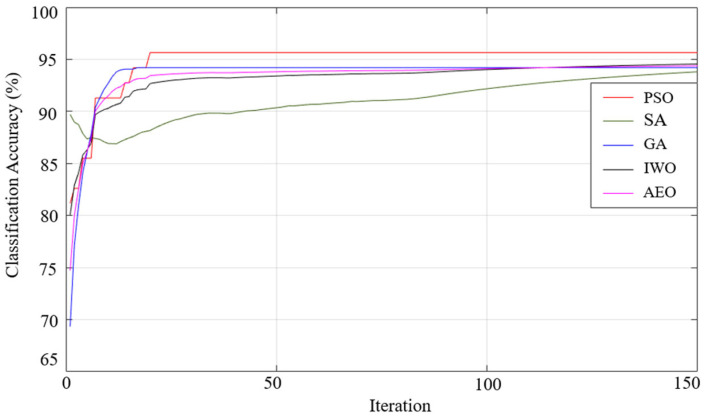
Performance of Gaussian membership function on classification accuracy.

**Figure 27 sensors-25-05271-f027:**
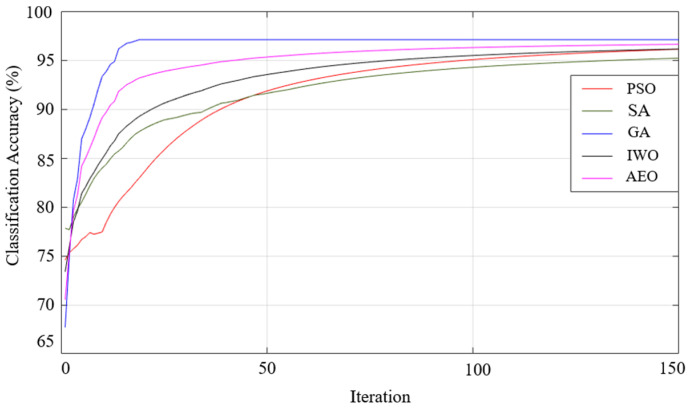
Performance of generalized bell-shaped membership function on classification accuracy.

**Figure 28 sensors-25-05271-f028:**
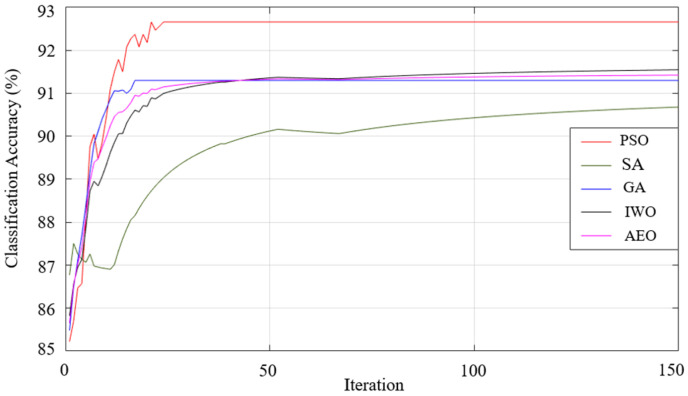
Performance of pi-shaped membership function on classification accuracy.

**Figure 29 sensors-25-05271-f029:**
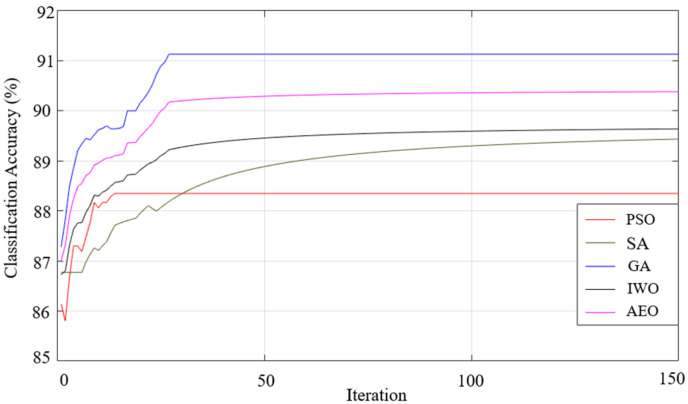
Performance of trapezoidal membership function on classification accuracy.

**Figure 30 sensors-25-05271-f030:**
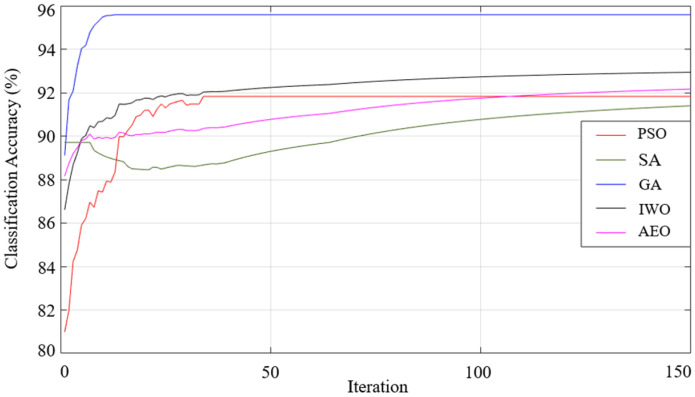
Performance of triangular membership function on classification accuracy.

**Figure 31 sensors-25-05271-f031:**
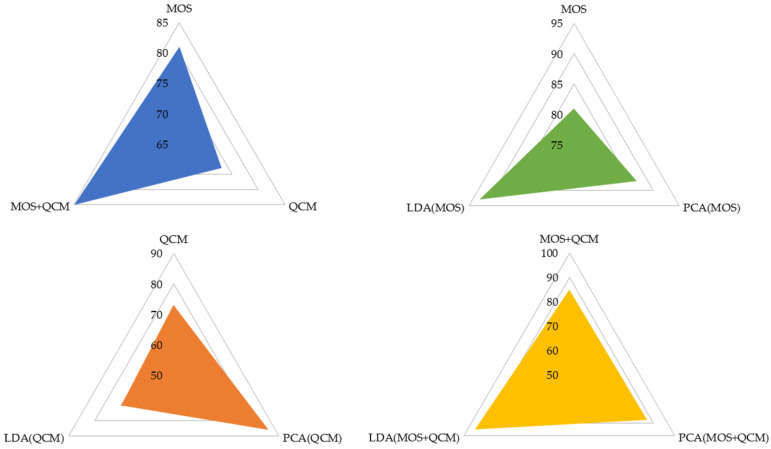
A graphical comparison of classification accuracies across different methods.

**Figure 32 sensors-25-05271-f032:**
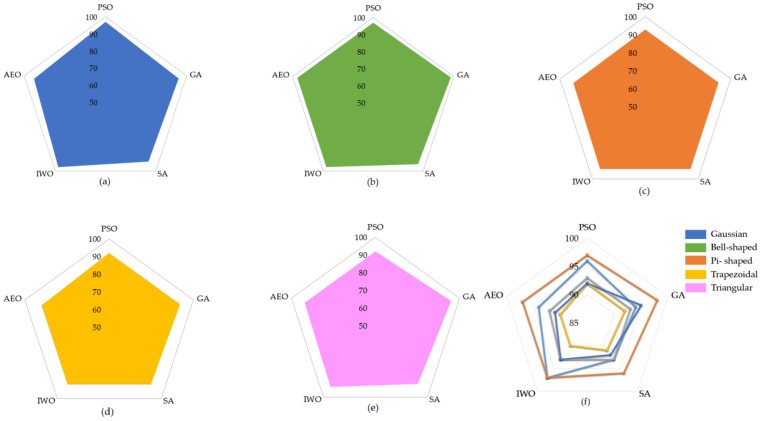
A comparative visualization of the classification performance across different methods: (**a**) Gaussian, (**b**) bell-shaped, (**c**) pi-shaped, (**d**) trapezoidal, (**e**) triangular, (**f**) mixed representation.

**Table 1 sensors-25-05271-t001:** The circuit and heater voltages of the MOS sensors.

	TGS8xx	TGS2xxx
$V_{C}$	24 V	5 V
$V_{H}$	5 V	5 V

**Table 2 sensors-25-05271-t002:** MOS sensors and load resistors.

TGS 813	TGS 816	TGS 826	TGS 832	TGS 2602	TGS 2610	TGS 2612	TGS 2620
6 kΩ	1 kΩ	2 kΩ	3 kΩ	1 kΩ	4.7 kΩ	2 kΩ	2 kΩ

**Table 3 sensors-25-05271-t003:** Detailed specifications for the MOS sensors.

Sensor Name	Target Gases	Cross-Sensitivity	Sensitivity Range	Sensor Resistance
TGS 813	Methanol, Propane, Ethanol, Hydrogen, Isobutane,	Carbon Monoxide	500–10,000 ppm	5–15 kΩ
TGS 816	Methane, Ethane, Hydrogen, Isobutane, Propane	Carbon Monoxide, Hydrogen peroxide	500–10,000 ppm	5–15 kΩ
TGS 826	Hydrogen, Isobutane Ethanol, Ammonia,	-	30–300 ppm	20–100 kΩ
TGS 832	R-12, Ethanol, R-22	Forane R134a	100–3000 ppm	4–40 kΩ
TGS 2602	Ammonia, Ethanol, Hydrogen Sulfide, Toluene	Hydrogen	1–30 ppm	10–100 kΩ
TGS 2610	Hydrogen, Iso-Propane Methane, Iso-Butane	Ethanol	300–10,000 ppm	0.68–6.8 kΩ
TGS 2612	Methane, Iso-Butane	Ethanol	500–12,500 ppm	0.68–6.8 kΩ
TGS 2620	Carbon Monoxide, Ethanol Iso-Butane, Hydrogen	Methane	50–500 ppm	1–5 kΩ

**Table 4 sensors-25-05271-t004:** Informative data of the volunteers.

	LC Patient (60 Person)	Healthy Volunteer (40 Person)
Age (Mean/St. deviation)	60.7/8	48.2/9
Gender (Female/Male)	13/47	10/30
Smokers/Non-smokers	0/60	20/20
Ex-smokers/Non-smokers	45/60	0/40

**Table 5 sensors-25-05271-t005:** The classification results.

Types of Features	Classification Algorithms					
	DT	L-SVM	Q-SVM	C-SVM	k-NN	RF
MOS	75.34 77.83-72.85-1.90	75.52 77-74.6-1.08	81.12 82.1-79.3-1.17	81.28 82-79.3-1.12	74.38 79.9-71.3-3.30	**81.54** **82.8-79.9-1.09**
PCA(MOS)	85.20 86.16-84.35-0.67	67.86 70.22-64.93-2.17	67.66 70.25-64.94-2.11	85.76 86.47-84.96-0.62	81.06 81.97-79.65-0.90	**87.16** **88.13-86.18-0.88**
LDA(MOS)	88.80 90.14-87.38-1.11	**93.20** **93.11-91.36-0.78**	92.60 92.96-91.72-0.45	91.56 92.84-90.25-1.06	90.10 90.81-89.47-0.51	90.04 90.87-89.16-0.80
QCM	66.82 70.75-64.22-2.70	64.32 64.83-64-0.43	71.96 74.07-70.15-1.46	69.66 73.45-66.56-2.51	70.50 72.28-68.92-1.33	**73.18** **75.14-70.75-1.77**
PCA(QCM)	74.96 76.92-73.75-1.25	67.86 70.27-64.91-2.17	67.66 70.23-64.96-2.11	82.22 84.33-79.37-1.93	**85.96** **87-84.95-0.89**	80.20 81-79-0.87
LDA(QCM)	67.30 68.61-65.72-1.30	67.86 70.24-64.95-2.17	67.66 70.28-64.96-2.11	68.92 70.12-68-0.78	66.76 70.45-63.36-2.95	**70.58** **71.27-69.94-0.56**
MOS + QCM	76.06 77.85-74-1.43	76.84 77.82-75.44-0.96	**85.26** **86.46-83.72-1.05**	85.18 86.11-84.16-0.79	75.38 76-74.93-0.48	82.24 82.92-81.45-0.58
PCA (MOS + QCM)	84.24 84.92-83.43-0.60	67.86 70.62-64.91-2.17	67.66 70.27-64.91-2.11	81.46 82.25-80.88-0.63	80.36 85.83-75.46-4.18	**87.56** **88.13-87.25-0.35**
LDA (MOS + QCM)	94.40 94.75-93.89-0.36	94.52 94.75-94.47-0.16	93.80 94.73-92.92-0.73	92.90 94.12-91.61-0.97	92.30 93.86-91.47-0.95	**94.58** **95.61-94.17-0.62**
PCA(MOS) + PCA(QCM)	83.40 85.26-82.08-1.33	67.86 70.25-64.92-2.17	70.02 70.75-69.270.56	83.10 83.74-82.28-0.60	**88.56** **88.83-88.26-0.25**	87.14 88.25-86.11-0.77
LDA(MOS) + LDA(QCM)	89.80 90.57-88.95-0.65	**93.08** **93.81-92-0.75**	92.60 93.81-92-0.73	90.74 91.47-89.65-0.80	90.60 90.57-89.6-0.35	90.92 91.4-90.2-0.54

**Table 6 sensors-25-05271-t006:** Rule table for classification with FL algorithm.

		**LD1**						
		$O_{1}$	$O_{2}$	${+}_{1}$	${+}_{2}$	${◊}_{1}$	${◊}_{2}$	${◊}_{3}$
**LD2**	$O_{1}$	O	O	O	O	O	◊	◊
	$O_{2}$	O	O	O	◊	◊	◊	◊
	${+}_{1}$	+	+	+	+	◊	◊	◊
	${+}_{2}$	+	+	+	+	+	◊	◊
	${◊}_{1}$	O	O	O	◊	◊	◊	◊
	${◊}_{2}$	O	O	O	◊	◊	◊	◊
	${◊}_{3}$	O	+	+	+	+	◊	◊

**Table 7 sensors-25-05271-t007:** Membership functions and their details.

**Membership Function**	**Graph and Equation of Membership Function**
Gaussian membership function	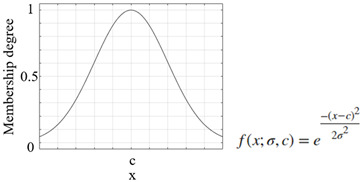
Generalized bell-shaped membership function	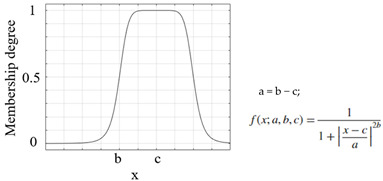
Triangular membership function	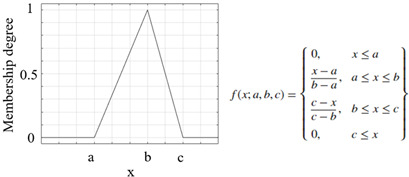
Trapezoidal membership function	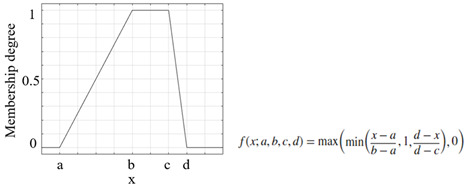
Pi-shaped membership function	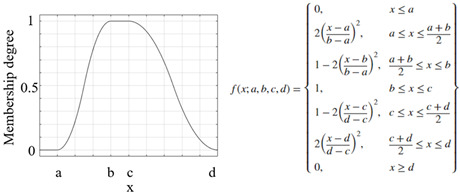

**Table 8 sensors-25-05271-t008:** Classification of optimization algorithms.

Algorithm	Tuning Parameters	Operators
GA	Crossover rate	Selection crossover rate, crossover mutation
PSO	Social acceleration coefficient, inertia weight, cognitive acceleration coefficient	Particle velocity update, particle position update
SA	Temperature	Annealing process
AEO	Energy transfer mechanism	Production, consumption, decomposition, reproduction
IWO	Invasive weed spread	Spectral spread, competitive deprivation

**Table 9 sensors-25-05271-t009:** Classification results based on the fuzzy logic method.

Membership Functions	Optimization Algorithms				
	PSO	GA	SA	IWO	AEO
Gaussian	95.69 98.07-92.80-2.41	94.81 97.95-92.07-2.27	92.95 97.36-87.22-3.77	97.27 98.81-94.71-1.66	93.82 97.06-91.18-2.18
Generalized Bell-Shaped	97.27 99.26-95.30-1.72	97.93 100-95.89-1.75	95.7 99.11-92.62-2.29	97.44 98.56-95.59-1.49	97.56 98.56-95.59-1.37
Pi	92.98 97.06-91.18-2.33	92.79 97.06-91.18-2.43	92.94 97.06-91.18-2.39	93.23 97.06-91.18-2.23	92.65 97.06-91.05-2.54
Trapezoidal	91.74 97.06-88.41-3.35	92.01 97.06-89.56-3.07	90.79 97.06-86.67-3.79	89.95 97.06-86.68-4.22	90.3 97.06-86.77-3.95
Triangular	92.3 95.59-89.55-2.22	94.81 95.59-92.07-2.27	91.62 95.59-89.71-2.51	92.7 97.06-9.71-2.77	91.47 95.59-89.56-2.63

**Table 10 sensors-25-05271-t010:** Hyperparameters of the optimization algorithms used in this study.

Algorithm	Population Size	Max. Iterations	Other Parameters
PSO	50	250	Inertia weight w = 0.7 (linearly decreasing 0.9 → 0.4), cognitive c1 = 1.5, social c2 = 1.5, velocity clamping = ±(range/2)
GA	60	250	Mutation rate = 0.05, Crossover rate = 0.8 (single/two-point), Selection = Tournament (size = 3), Elitism = 2 individuals
SA	-	250	Initial temperature T0 chosen s.t. initial acceptance ≈ 0.8; Cooling: geometric $T_{k+1}=$ α$T_{k}$ with α = 0.995; Neighbor perturbation = Gaussian step, σ_initial = 0.1 × variable_range, σ_final = 1 × 10^−4^ × range
IWO	50 (Initial seeds)	250	Min seeds = 2; Max seeds = 6; σ_initial = 3.0; σ_final = 0.01; Reproduction rule: seeds ∝ fitness (linear)
AEO	50	250	Producer ratio = 0.5; Consumer ratio = 0.3; Decomposer prob. = 0.2; Migration rate = 0.1; Environmental Factor (EF) = 0.5

## Data Availability

The data presented in this study are not publicly available at this time because they are part of ongoing research. The data may be made available by the authors upon reasonable request once the related studies are completed.

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
