# Peer review of "Enhanced Lung Cancer Classification Accuracy via Hybrid Sensor Integration and Optimized Fuzzy Logic-Based Electronic Nose"

_sensors, 2025, doi:10.3390/s25175271_

Round 1
Reviewer 1 Report
Comments and Suggestions for Authors
This paper presents a hybrid electronic nose system integrating eight metal oxide semiconductor (MOS) and fourteen quartz crystal microbalance (QCM) sensors to classify lung cancer from breath samples, achieving 97.93% accuracy using an optimized fuzzy logic classifier enhanced by nature-inspired algorithms. The study demonstrates that combining multidimensional sensor data with intelligent fuzzy logic optimization offers a promising noninvasive approach for early lung cancer diagnosis.
Here, authors can find some comments to improve their paper:
1. Check page 25.
2. Clarify if the cancer patients’ stages and histological subtypes were balanced across the dataset to avoid bias.
3. Were environmental VOC contamination, dietary restrictions, or time-of-day effects controlled during breath sampling to minimize confounding?
4. The authors extract max, variance, mean, kurtosis, skewness, and gradient features. Justify why these were selected over other time-series features, such as wavelet coefficients or frequency domain features.
5. Provide quantitative justification or optimization for the rule table for the fuzzy classifier design, or it was purely heuristic?
6. Improve your literature review by adding new references in the field of E nose, such as DOI: 10.1016/j.snb.2022.131418.
7. Were MOS and QCM sensors exposed simultaneously to the same breath sample stream, and was there any delay or temporal mismatch that could affect hybrid data fusion?
Author Response
Authors’ Response to Reviews
The authors would like to thank the reviewers and editor for their valuable comments. The recommended changes and detailed explanations are as follow. All changes in the manuscript are indicated with red color.
Reviewer 1 :
- Check page 25.
*The extra part on page 25 was deleted.
- Clarify if the cancer patients’ stages and histological subtypes were balanced across the dataset to avoid bias.
* Information on the stages and histological subtypes of cancer patients participating in the study is provided in Figure 12 on page 8.
- Were environmental VOC contamination, dietary restrictions, or time-of-day effects controlled during breath sampling to minimize confounding?
* To participate in the experiments, individuals were instructed to fast overnight and to provide breath samples in a fasted state. They were also required to refrain from smoking for at least two hours prior to sample collection. All breath samples were collected between 8:00 and 11:00 a.m. to maintain consistency from the beginning of the study. Furthermore, a timing method was implemented to minimize environmental VOC contamination. These procedures are described in detail in the final paragraph of page 11.
- The authors extract max, variance, mean, kurtosis, skewness, and gradient features. Justify why these were selected over other time-series features, such as wavelet coefficients or frequency domain features.
* In the literature on electronic nose (e-nose) systems, the types of features utilized are generally classified into three main categories. The first approach involves the direct extraction of segmented features from the raw sensor response curves. These features typically include maximum values, integrals, differential values, first- and second-order derivatives, as well as adsorption slopes—particularly the maximum slope observed within specific time intervals. The second approach is based on curve fitting, in which sensor response curves are modeled using predefined mathematical functions, and the resulting fitting parameters are used as distinctive features. The third approach relies on signal transformation techniques, with fast Fourier transform (FFT) and discrete wavelet transform (DWT) being among the most commonly employed tools for this purpose [1].
In e-nose systems, the temporal variation of sensor signals is directly influenced by the volatile organic compounds (VOCs) to which the sensors are exposed. Therefore, analyzing this temporal change in detail and extracting features that effectively capture differences in the data is essential.
In the present study, a detailed analysis of the temporal behavior of the signals was conducted. Based on this analysis, six time-domain features—maximum value, variance, mean, kurtosis, skewness, and gradient—were selected for use. Previous studies have also demonstrated the effectiveness of these features in capturing temporal variations in sensor data [2].
Frequency-based features were not employed, as the sensor signals in this study exhibited slow variations over time, rendering frequency-domain information insufficiently discriminative. Additionally, curve-fitting-based features were excluded due to the additional computational burden associated with model fitting procedures.
[1]. Yan, J.; Guo, X.; Duan, S.; Jia, P.; Wang, L.; Peng, C.; Zhang, S. Electronic Nose Feature Extraction Methods: A Review. Sensors 2015, 15, 27804-27831.
[2]. Llobet, E.; Brezmes, J.; Vilanova, X.; Sueiras, J.E.; Correig, X. Qualitative and quantitative analysis of volatile organic compounds using transient and steady-state responses of a thick-film tin oxide gas sensor array. Sens. Actuators B Chem. 1997, 41, 13–21.
- Provide quantitative justification or optimization for the rule table for the fuzzy classifier design, or it was purely heuristic?
* To construct the rule table, the membership functions were initially placed on the scatter plot. During this initialization, the scatter plots for all classes were divided into equal segments along both the LD1 and LD2 axes, considering the distribution of the data. Based on this evaluation, two membership functions were assigned for each of the HnS and HS classes, and three for the cancer patient class. The initial boundaries of these membership functions are shown in Figure 1. Due to the visual complexity of this figure, it is referred to as Figure 16 throughout the article for clarity.
After the initial placement, the rule table was developed by analyzing the intersection regions of the membership functions depicted in Figure 1. For each intersection region, the rule was assigned to the class that contained the highest number of data points within that region.
Furthermore, the following clarification was added to page 18 of the article: " The rule table was constructed by analyzing the intersection sets of the membership func-tions. For each intersection, the rule was assigned to the class that had the largest number of data points within that region."
Figure 1. The initial placement of fuzzy sets on the scatter plot
- Improve your literature review by adding new references in the field of E nose, such as DOI: 10.1016/j.snb.2022.131418.
* The recommended publication has been added to the manuscript and is now listed as reference [27].
- Were MOS and QCM sensors exposed simultaneously to the same breath sample stream, and was there any delay or temporal mismatch that could affect hybrid data fusion?
* In the experimental setup, both the MOS and QCM sensor chambers are positioned in parallel and placed equidistant from the vacuum pump that delivers the breath samples. This configuration ensures that the samples reach both sensor chambers simultaneously. As illustrated in Figure 4 on page 5, the sensors are exposed to the odors at the same time and generate responses concurrently.

Reviewer 2 Report
Comments and Suggestions for Authors
This manuscript reports on the development of a hybrid electronic nose system integrating eight metal oxide semiconductor (MOS) and fourteen quartz crystal microbalance (QCM) sensors for noninvasive lung cancer diagnosis via breath analysis. The study combines multidimensional sensor data with a fuzzy logic (FL) classifier optimized through nature-inspired algorithms, achieving a reported classification accuracy of 97.93%, which surpasses several conventional machine learning methods. The integration of dual sensor modalities with an optimized fuzzy inference approach is timely and relevant to the field of breath-based disease diagnostics. Nevertheless, the manuscript has several critical shortcomings that must be addressed before it can be considered for publication. These include insufficient clinical and statistical rigor, incomplete characterization of the sensing hardware, limited discussion of confounding factors, and a lack of benchmarking against state-of-the-art computational models. Given these concerns, I recommend a major revision of the manuscript before it can be considered for publication.
The detailed comments are provided below:
- The study is based on 338 breath measurements from 100 participants (60 lung cancer patients and 40 healthy volunteers). While promising, this constitutes a relatively small dataset for a diagnostic study, particularly when subdividing by cancer stage, histological type, and smoking status. The authors are strongly encouraged to provide statistical power analysis or confidence intervals for the reported classification accuracies. The lung cancer cohort is predominantly composed of ex-smokers (45 out of 60 participants), whereas the healthy group contains a fixed 1:1 ratio of current smokers to non-smokers. This uneven distribution of smoking status between groups introduces potential confounding effects, as the classifier may inadvertently capture differences related to smoking history or cessation rather than lung cancer–specific metabolic signatures in the breath. Given that volatile organic compound (VOC) profiles are known to be strongly influenced by both active smoking and residual metabolic effects in ex-smokers, the current design makes it difficult to disentangle whether the reported classification performance is driven by cancer-related biomarkers or by smoking-related patterns. To address this concern, it is recommended that the authors conduct a stratified performance analysis based on smoking status (current smokers, ex-smokers, and non-smokers) to evaluate whether the model maintains diagnostic accuracy independent of this variable. If such an analysis is not feasible due to sample size constraints, the authors should explicitly acknowledge this limitation in the discussion and provide a rationale for how it may impact the generalizability of the findings.
- The hybrid use of MOS and QCM sensors is a key innovation of this work, but the manuscript lacks sufficient hardware characterization. It is necessary to provide the full specifications of each MOS sensor, including sensitivity range, baseline resistance, and known cross-sensitivities, as well as detailed QCM calibration curves that link frequency shifts to mass loading or VOC concentration and include the Sauerbrey constant with the relevant equations. In addition, stability and drift tests over repeated measurements should be presented to demonstrate the long-term reproducibility of the hybrid array. Environmental effects (temperature, humidity) are not addressed, although they are known to strongly influence both MOS and QCM responses. Please clarify whether compensation mechanisms were implemented or controlled experimental conditions were maintained.
- It should be noted that the presentation format of the data in Table 2 requires attention. The significant figures are inconsistent across different entries, which may affect the clarity and precision of the reported results. Furthermore, the use of commas instead of decimal points for numerical values does not conform to standard scientific formatting in English-language journals. It is recommended that the table be revised to ensure uniform significant figures and that decimal points be used consistently throughout.
- The feature set is dominated by static statistical descriptors (mean, variance, skewness, kurtosis), while the transient kinetics of sensor responses are largely ignored. It is recommended to extract dynamic features such as response and recovery times (T90, T10), adsorption/desorption slopes, or apply time-series modeling approaches. PCA and LDA are applied, but the manuscript does not report the variance explained ratio or eigenvalue distribution. Providing these metrics would strengthen the justification for dimensionality reduction choices.
- The manuscript does not sufficiently link the selected MOS/QCM target gases to known volatile biomarkers of lung cancer. Please discuss the specific VOCs each sensor is intended to detect and how these correlate with established metabolic pathways in lung cancer patients. Breath sampling via Tedlar bags and the subsequent dry-air purge introduce potential dilution and carryover effects, which could significantly influence the accuracy and consistency of the sensor readings. These effects can lead to variability in the sensor responses due to incomplete clearing of residual gases, potentially skewing the results and impacting the reproducibility of the study. To address this concern, it is essential for the authors to provide a detailed quantification of the sample volume used in the collection process, including the volume of breath extracted from each participant and the capacity of the Tedlar bags. In addition, the manuscript should report the residual gas clearance efficiency, which refers to the effectiveness of the dry-air purge in removing any residual gases from previous samples.
- The authors are also encouraged to describe any standardization protocols implemented during the sampling process to minimize variability, such as the use of controlled flow rates, consistent purge times, or temperature regulation. Providing this information will help to assess the precision of the sampling technique and its impact on the overall experimental results, ensuring that any observed differences are truly due to the targeted biomarkers rather than procedural inconsistencies.
- The introduction would benefit from a more comprehensive comparison to other multimodal e-nose studies for lung cancer detection, highlighting how this work differs from previous MOS-only or QCM-only systems. It is recommended that the manuscript discuss the potential for integrating this hybrid system into clinical workflows, including considerations of cost, portability, and standardization challenges to highlight its translational relevance.
- The manuscript notes that the dataset is not publicly available due to ongoing research. Consider releasing a subset of anonymized data or providing a detailed data dictionary to facilitate reproducibility. It would be beneficial to include a summary of all hyperparameters used in the FL optimization process, such as population sizes, mutation rates, and iteration counts, in a supplementary table to facilitate replication.
- The FL classifier is optimized with five nature-inspired algorithms (GA, PSO, SA, IWO, AEO). While the reported 97.93% accuracy is impressive, the study lacks benchmarking against contemporary deep learning models (e.g., CNN, LSTM, or multimodal fusion networks), which are now standard in advanced e-nose research. Including such comparisons would contextualize the benefits and trade-offs of the FL approach. The creation of the fuzzy rule base appears to rely heavily on expert-defined initial placements. Please clarify whether any automated rule-learning strategies were tested and provide confusion matrices or ROC/AUC curves to substantiate the classifier’s performance beyond mean accuracy. There are many studies about the deep learning methods of the E-nose for gas detection. I suggest the authors have some review or discussion of these articles if appropriate, e.g., ACS Sens. 2025, 10, 2531–2541.
Author Response
Authors’ Response to Reviews
The authors would like to thank the reviewers and editor for their valuable comments. The recommended changes and detailed explanations are as follow. All changes in the manuscript are indicated with red color.
Reviewer 1 :
1 (a). The study is based on 338 breath measurements from 100 participants (60 lung cancer patients and 40 healthy volunteers). While promising, this constitutes a relatively small dataset for a diagnostic study, particularly when subdividing by cancer stage, histological type, and smoking status. The authors are strongly encouraged to provide statistical power analysis or confidence intervals for the reported classification accuracies.
* In order to address this comment, three review articles from the literature on lung cancer detection using electronic nose technology have been examined.
In the first review article, an analysis of 40 studies reported that the average number of lung cancer patients was 49.92 (maximum: 252, minimum: 6), while the average number of healthy volunteers was 69.7 (maximum: 411, minimum: 6) [1]. The second review article, which also analyzed 40 studies, indicated that the average number of lung cancer patients was 47.42 (maximum: 114, minimum: 21), whereas the average number of healthy volunteers was 54.10 (maximum: 114, minimum: 10) [2]. In the third review article, covering 27 studies, the average number of lung cancer patients was reported as 73.03 (maximum: 252, minimum: 6), while the average number of healthy volunteers was 68.44 (maximum: 223, minimum: 5) [3].
Based on these findings, the participation of 100 volunteers (60 lung cancer patients and 40 healthy individuals) in the present study is considered sufficient when compared to similar studies in the literature. However, increasing the number of experiments would undoubtedly enhance the statistical accuracy and reliability of the results.
[1] Vadala, Rohit, et al. "A review on electronic nose for diagnosis and monitoring treatment response in lung cancer." Journal of Breath Research 17.2 (2023): 024002.
[2] C. Baldini, L. Billeci, F. Sansone, R. Conte, C. Domenici, and A. Tonacci, “Electronic Nose as a Novel Method for Diagnosing Cancer: A Systematic Review,” Biosensors, vol. 10, no. 8, p. 84, Jul. 2020, doi: https://doi.org/10.3390/bios10080084.
[3] M. H. M. C. Scheepers, Z. Al-Difaie, L. Brandts, A. Peeters, B. van Grinsven, and N. D. Bouvy, “Diagnostic Performance of Electronic Noses in Cancer Diagnoses Using Exhaled Breath,” JAMA Network Open, vol. 5, no. 6, p. e2219372, Jun. 2022, doi: https://doi.org/10.1001/jamanetworkopen.2022.19372.
1(b). The lung cancer cohort is predominantly composed of ex-smokers (45 out of 60 participants), whereas the healthy group contains a fixed 1:1 ratio of current smokers to non-smokers. This uneven distribution of smoking status between groups introduces potential confounding effects, as the classifier may inadvertently capture differences related to smoking history or cessation rather than lung cancer–specific metabolic signatures in the breath. Given that volatile organic compound (VOC) profiles are known to be strongly influenced by both active smoking and residual metabolic effects in ex-smokers, the current design makes it difficult to disentangle whether the reported classification performance is driven by cancer-related biomarkers or by smoking-related patterns. To address this concern, it is recommended that the authors conduct a stratified performance analysis based on smoking status (current smokers, ex-smokers, and non-smokers) to evaluate whether the model maintains diagnostic accuracy independent of this variable. If such an analysis is not feasible due to sample size constraints, the authors should explicitly acknowledge this limitation in the discussion and provide a rationale for how it may impact the generalizability of the findings.
* We acknowledge the reviewer’s concern regarding the potential confusion caused by the uneven distribution of smoking status among groups. However, in our study, the use of two different types of gas sensors, combined with a meticulous feature extraction process to identify the most discriminative parameters between classes, allowed the classifiers to effectively differentiate the data. For instance, as shown in the confusion matrix in Figure 15(c), breath samples from lung cancer patients were successfully distinguished from those of healthy smokers. Out of 218 data points belonging to the lung cancer class, only 2 were misclassified as healthy smokers. Similarly, out of 55 data points in the healthy smoker class, only 2 were misclassified as lung cancer. These results demonstrate that the potential negative impact of smoking status on classifier performance has been substantially minimized in our work.
2(a).The hybrid use of MOS and QCM sensors is a key innovation of this work, but the manuscript lacks sufficient hardware characterization. It is necessary to provide the full specifications of each MOS sensor, including sensitivity range, baseline resistance, and known cross-sensitivities, as well as detailed QCM calibration curves that link frequency shifts to mass loading or VOC concentration and include the Sauerbrey constant with the relevant equations.
* The basic measuring circuit of the MOS sensors has been added as Figure 5 on page 5. In this figure, and represent the circuit and heater voltages, respectively. Circuit and heater voltages of the MOS sensors used in this study are listed in Table 3. and represent the load resistance connected in series with the sensor and the voltage value obtained across this resistance, respectively. The detailed information about this circuit and sensors have been added as Table 1, 2 and 3 on page 6.
Figure 4. Basic measuring circuit of MOS sensors.
Table 1. The circuit and heater voltages of the MOS sensors
|
TGS8xx |
TGS2xxx |
24 V |
5 V |
|
5 V |
5 V |
Table 2. MOS sensors, and load resistors.
TGS 813 |
TGS 816 |
TGS 826 |
TGS 832 |
TGS 2602 |
TGS 2610 |
TGS 2612 |
TGS 2620 |
6k Ω |
1k Ω |
2k Ω |
3k Ω |
1k Ω |
4.7k Ω |
2k Ω |
2k Ω |
Table 3. Detailed specifications for the MOS sensors
Sensor Name |
Target Gases |
Cross-Sensitivity (Interfering Gases) |
Sensitivity range |
Sensor resistance |
TGS 813 |
Methanol, Ethanol, Propane, Isobutane, Hydrogen |
Carbon Monoxide |
500–10 000 ppm |
5kΩ-15 kΩ |
TGS 816 |
Methane, Ethane, Propane, Isobutane, Hydrogen |
Carbon Monoxide, Hydrogen peroxide |
500–10 000 ppm |
5kΩ-15 kΩ |
TGS 826 |
Isobutane, Hydrogen, Ammonia, Ethanol |
- |
30–300 ppm |
20kΩ-100 kΩ |
TGS 832 |
R-12, Ethanol, R-22 |
Forane R134a |
100–3 000 ppm |
4kΩ-40 kΩ |
TGS 2602 |
Ammonia, Ethanol, Hydrogen Sulfide, Toluene |
Hydrogen |
1–30 ppm |
10kΩ-100 kΩ |
TGS 2610 |
Hydrogen, Methane, Iso-Butane, Iso-Propane |
Ethanol |
300–10 000 ppm |
0.68kΩ-6.8 kΩ |
TGS 2612 |
Methane, Iso-Butane |
Ethanol |
500–12 500 ppm |
0.68kΩ-6.8 kΩ |
TGS 2620 |
Carbon Monoxide, Iso-Butane, Hydrogen, Ethanol |
Methane |
50–500 ppm |
1kΩ-5kΩ |
* The detailed information about QCM sensors has been added as follows on page 6-7. Quartz crystal microbalance (QCM) sensors are another type of gas sensor used in this. These sensors feature a quartz disk situated between two parallel gold electrodes. Quartz, being a piezoelectric material, experiences mechanical deformation when an alternating current is applied to the electrodes. If the current's frequency matches the QCM sensor's resonant frequency, a standing wave forms within the resonator, which is linked to the sensor's mass. The sensors are coated with a thin film that reacts to specific gases. When the target gas particles are absorbed by this film, the sensor's resonant frequency shifts. Under certain conditions, known as small load assumptions, the change in frequency can be roughly measured and linked to the alteration in mass by employing the well-established Sauerbrey’s equation:
where represents the change in frequency (Hz), denotes the sensitivity factor of the employed quartz crystal (for instance, 56.6 Hz cm2/μg for a 5 MHz AT-cut quartz crystal at room temperature), and signifies the change in mass per unit area (Rodahl & Kasemo, 1996). The sensitivity factor can be calculated as follows:
where represents the shear modulus of quartz (2.947xg/cm), denotes the quartz density (2.648 g/), n is the harmonic number at which the crystal is driven, and is the fundamental resonant frequency of the quartz crystal (Dixon, 2008).
The dissipation factor is determined by taking the reciprocal of the resonance quality factor.
The formula, where w represents the bandwidth, measures the system's damping. It can also be calculated as (Dixon, 2008):
where represents the frequency change in Hertz, and τ denotes the decay constant of the quartz resonator. This can be interpreted as the ratio of the energy lost during each oscillation to the product of a constant and the total energy stored within the system, essentially comparing dissipated energy to conserved energy (Dixon, 2008):
Δf is linked to the quantity of the sample that is adsorbed or desorbed, while D is associated with the sample's stiffness and viscoelastic properties.
This frequency change is measured and relayed to a computer via an appropriate electronic circuit. By analyzing this frequency shift, valuable insights about the target gases can be obtained. A basic working principle of quartz crystal microbalance sensor has been added as Figure 5 on page 7.
Figure 5. Basic working principle of quartz crystal microbalance sensor
2(b). In addition, stability and drift tests over repeated measurements should be presented to demonstrate the long-term reproducibility of the hybrid array.
During the experiments, the signals obtained from the sensors were analyzed regularly. The uninterrupted operation and regular cleaning of the sensors with dry air prevented significant amplitude changes in the sensor signals and no drift was observed in the signals obtained from the sensors.No stability and drift tests was performed in this study.
2(c). Environmental effects (temperature, humidity) are not addressed, although they are known to strongly influence both MOS and QCM responses. Please clarify whether compensation mechanisms were implemented or controlled experimental conditions were maintained.
* In this study, constant temperature and humidity levels were maintained in the laboratory room where the experiments were conducted by using an air conditioner.
- It should be noted that the presentation format of the data in Table 2 requires attention. The significant figures are inconsistent across different entries, which may affect the clarity and precision of the reported results. Furthermore, the use of commas instead of decimal points for numerical values does not conform to standard scientific formatting in English-language journals. It is recommended that the table be revised to ensure uniform significant figures and that decimal points be used consistently throughout.
* We appreciate the reviewer’s observation regarding the presentation format in Table 2. In response, we have revised the table to ensure that all numerical values are presented with a uniform number of significant figures. Specifically, all data are now given with two decimal places, and decimal points are used instead of commas to conform to standard scientific formatting in English-language journals. This change improves both the clarity and precision of the reported results.
4(a). The feature set is dominated by static statistical descriptors (mean, variance, skewness, kurtosis), while the transient kinetics of sensor responses are largely ignored. It is recommended to extract dynamic features such as response and recovery times (T90, T10), adsorption/desorption slopes, or apply time-series modeling approaches.
* In the study, in addition to features such as mean, variance, skewness, and kurtosis, transient response values were also used as features. In the article, these features were simply presented as “gradient of the data across specified temporal intervals.” In this study, the data were carefully examined prior to feature extraction, and in addition to static statistical descriptors, the following features that best differentiate between classes were also used:
- The area between TGS826 and TGS832 sensor data in the interval 130-170 seconds
- The area between TGS813 and TGS2620 sensor data in the interval 130-170 seconds
- The area between TGS880 and TGS2610 sensor data in the interval 130-170 seconds
- The slope of all sensor data in the interval 130-145 seconds
- The slope of all sensor data in the interval 145-170 seconds
. The slope of sensor data in the interval 170-200 seconds
- The slope of all sensor data in the interval 200-215 seconds
- The slope of sensor data in the interval 200-230 seconds
. The area under the all sensor data in the interval 145-215 seconds
These features have been added to page 13 and 14.
4(b). PCA and LDA are applied, but the manuscript does not report the variance explained ratio or eigenvalue distribution. Providing these metrics would strengthen the justification for dimensionality reduction choices.
* The percentage ratios of the eigenvalues required for selecting the planes in the LDA method, together with those corresponding to the principal components, are shown in Figure 18. This figure has been added to page 15. Upon reviewing the classification outcomes presented in Figures 20 and 21, it is evident that the feature matrices generated through LDA are classified with higher accuracy than those derived from PCA.
Figure 18. Percentage contribution of eigenvalues for PCA and LDA
5(a). The manuscript does not sufficiently link the selected MOS/QCM target gases to known volatile biomarkers of lung cancer. Please discuss the specific VOCs each sensor is intended to detect and how these correlate with established metabolic pathways in lung cancer patients.
* The distinctive VOCs present in the breath of lung cancer patients are detailed extensively in the introduction on page 3. The target gases for the MOS sensors used in the study are also listed in Table 3, which is included in the article. Based on this information, although gas sensors may not have high sensitivity to all VOCs originating from lung cancer, they are nevertheless capable of detecting a significant number of them. Furthermore, the MOS sensors used in this study have been successfully utilized in many studies found in the literature [4, 5, 6]. The QCM sensors used in this study were obtained from TÜBİTAK MAM. A list of desired VOC was provided to the company, which produced QCM sensors with these VOCs in mind.
[4]. Li, W., Liu, H., Xie, D., He, Z., & Pi, X. (2017). Lung cancer screening based on type-different sensor arrays. Scientific reports, 7(1), 1969.
[5]. Guo, D., Zhang, D., Li, N., Zhang, L., & Yang, J. (2010). A novel breath analysis system based on electronic olfaction. IEEE transactions on biomedical engineering, 57(11), 2753-2763.
[6]. Marzorati, D., Mainardi, L., Sedda, G., Gasparri, R., Spaggiari, L., & Cerveri, P. (2021). MOS sensors array for the discrimination of lung cancer and at-risk subjects with exhaled breath analysis. Chemosensors, 9(8), 209.
5(b). Breath sampling via Tedlar bags and the subsequent dry-air purge introduce potential dilution and carryover effects, which could significantly influence the accuracy and consistency of the sensor readings. These effects can lead to variability in the sensor responses due to incomplete clearing of residual gases, potentially skewing the results and impacting the reproducibility of the study. To address this concern, it is essential for the authors to provide a detailed quantification of the sample volume used in the collection process, including the volume of breath extracted from each participant and the capacity of the Tedlar bags.
* During the breath sample collection process, a single Tedlar bag was used for each volunteer to collect their breath sample. Please note that the Tedlar bags used were not cleaned with dry air to avoid any misunderstanding. The breath samples were collected until the 5-liter Tedlar bags were completely filled. Once the Tedlar bag was full and no more breath samples could be added, the breath collection process was concluded. Two experiments were conducted with the breath samples in each Tedlar bag. A detailed description of the breath collection process has been added to page 11 of the manuscript.
5(c). In addition, the manuscript should report the residual gas clearance efficiency, which refers to the effectiveness of the dry-air purge in removing any residual gases from previous samples.
* The e-nose was cleaned with dry air for 130 s before each experiment and for 140 s after each experiment. These durations were determined based on observations made during the e-nose development phase. The periods when the sensor responses stabilized were identified, and it was observed that the sensor cleaning was effectively conducted during these periods. Graphs related to two randomly consecutive experiments can be seen as follows. These graphs show that the level to which the sensor responses decreased before and after the experiments, following the cleaning process, demonstrates that the sensors were successfully cleaned.
Figure. MOS sensor graphs obtained after two consecutive experiments
- The authors are also encouraged to describe any standardization protocols implemented during the sampling process to minimize variability, such as the use of controlled flow rates, consistent purge times, or temperature regulation. Providing this information will help to assess the precision of the sampling technique and its impact on the overall experimental results, ensuring that any observed differences are truly due to the targeted biomarkers rather than procedural inconsistencies.
* In this study, constant temperature and humidity levels were maintained in the laboratory room where the experiments were conducted using an air conditioner. The pressure control valve of the dry air cylinder was maintained at a constant angle at all times to ensure that the entire system was always cleaned with dry air at the same flow rate. A constant supply voltage was applied to the vacuum pump that delivered the breath samples to the sensors, ensuring that the breath samples were delivered to the sensors at the same speed during all experiments. This information can be found in the Materials and Methods section on page 9 of the article.
- The introduction would benefit from a more comprehensive comparison to other multimodal e-nose studies for lung cancer detection, highlighting how this work differs from previous MOS-only or QCM-only systems. It is recommended that the manuscript discuss the potential for integrating this hybrid system into clinical workflows, including considerations of cost, portability, and standardization challenges to highlight its translational relevance.
* The paragraph below has been included in the Introduction section on page 4 of the revised manuscript.
A review of studies in the literature on lung cancer detection using electronic noses reveals notable differences in diagnostic performance depending on the type of sensor employed. In 11 studies where only metal oxide semiconductor (MOS) sensors were used, the mean sensitivity, specificity, and accuracy were 89% (range: 78–95%), 80% (range: 33–100%), and 87% (range: 73–97%), respectively. In contrast, in 8 studies that utilized only frequency-based sensors, the mean sensitivity, specificity, and accuracy were 85% (range: 76–98%), 89% (range: 75–100%), and 89% (range: 75–100%), respectively. These findings suggest that while MOS-based systems tend to achieve slightly higher sensitivity, frequency-based systems may offer superior specificity, highlighting the importance of sensor selection in the design of electronic nose systems for clinical diagnostics.
* The paragraph below has been included in the discussion section on page 24-25 of the revised manuscript.
The developed e-nose is currently in an experimental setup. However, efforts are ongoing to create the smallest possible prototype of this experimental setup. It is also planned to include a section in the e-nose under development that will allow breath samples to be collected without using a Tedlar bag. Thus, the e-nose can become a device that can easily be carried by hand. However, the only limiting factor is the dry air cylinder. Although the e-nose circuit can be miniaturized and made portable, a dry air cylinder is still required at the location where the experiments are conducted.
8(a). The manuscript notes that the dataset is not publicly available due to ongoing research. Consider releasing a subset of anonymized data or providing a detailed data dictionary to facilitate reproducibility.
Rather than a subset of anonymized data, a detailed data dictionary has been provided as follows.
Data Dictionary – Breath Sample Sensor Dataset
General Information
Dataset Description:
This dataset contains time-series measurements obtained from 8 gas sensors during breath sample analysis. Each row corresponds to one sampled data point in chronological order.
Total Rows: ~34,500 (number of recorded sample points)
Total Columns: 9
Unit of Measurement: Volt (V)
Sampling Context: Measurements were collected during experimental procedures using the developed electronic nose system.
Column Name |
Description |
Data Type |
Unit |
Example Value |
Sample_Index |
Sequential number indicating the order of acquisition of each sample point. |
Integer |
— |
10235 |
Sensor_1 |
Output voltage from gas sensor #1. |
Float |
Volt |
1.254 |
Sensor_2 |
Output voltage from gas sensor #2. |
Float |
Volt |
1.487 |
Sensor_3 |
Output voltage from gas sensor #3. |
Float |
Volt |
0.932 |
Sensor_4 |
Output voltage from gas sensor #4. |
Float |
Volt |
2.145 |
Sensor_5 |
Output voltage from gas sensor #5. |
Float |
Volt |
0.756 |
Sensor_6 |
Output voltage from gas sensor #6. |
Float |
Volt |
1.638 |
Sensor_7 |
Output voltage from gas sensor #7. |
Float |
Volt |
2.011 |
Sensor_8 |
Output voltage from gas sensor #8. |
Float |
Volt |
1.223 |
Notes:
Sensor readings are raw values without any normalization or scaling applied.
The Sample_Index is unique and monotonically increasing for the duration of the measurement process.
Measurements were collected under consistent experimental conditions to ensure comparability.
8(b). It would be beneficial to include a summary of all hyperparameters used in the FL optimization process, such as population sizes, mutation rates, and iteration counts, in a supplementary table to facilitate replication.
* Hyperparameters of the optimization algorithms used in this study has been added as Table 10 on page 22
9(a). The FL classifier is optimized with five nature-inspired algorithms (GA, PSO, SA, IWO, AEO). While the reported 97.93% accuracy is impressive, the study lacks benchmarking against contemporary deep learning models (e.g., CNN, LSTM, or multimodal fusion networks), which are now standard in advanced e-nose research. Including such comparisons would contextualize the benefits and trade-offs of the FL approach.
* We thank the reviewer for this insightful comment. We acknowledge that benchmarking against contemporary deep learning models (e.g., CNN, LSTM, or multimodal fusion networks) is an important aspect in advanced E-nose research. Although such implementations are beyond the current scope of this study, we plan to explore and compare these models in our future work.
9(b). The creation of the fuzzy rule base appears to rely heavily on expert-defined initial placements. Please clarify whether any automated rule-learning strategies were tested and provide confusion matrices or ROC/AUC curves to substantiate the classifier’s performance beyond mean accuracy.
* To establish the rules, membership functions were first placed as explained on page 18, and then, using the procedures given in Figure 23, the positions of the membership functions were optimized to achieve the highest classification accuracy.
The related confusion matriz has been added as Figure 24 on Page 21.
9(c). There are many studies about the deep learning methods of the E-nose for gas detection. I suggest the authors have some review or discussion of these articles if appropriate, e.g., ACS Sens. 2025, 10, 2531–2541.
* We thank the reviewer for this insightful comment. We acknowledge that benchmarking against contemporary deep learning models (e.g., CNN, LSTM, or multimodal fusion networks) is an important aspect in advanced E-nose research. Although such implementations are beyond the current scope of this study, we plan to explore and compare these models in our future work. In response to the reviewer’s suggestion, we have expanded the references in Introduction section to include recent studies employing deep learning approaches for E-nose-based gas detection, providing additional context for our research.

Reviewer 3 Report
Comments and Suggestions for Authors
This research is useful and suitable for publication in a journal. However, there are some minor issues that need to be improved:
1) Authors should present a flowchart or diagram showing the overall research framework or process.
2) Authors should clearly detail the dataset used in tables to facilitate readers' study and research development.
3) Has this research been developed into an application? If so, this should be further presented to complement the presented hardware.
4) If the research results can clearly show the sample results obtained from classification, this will further demonstrate the contribution of the research.
5) Authors should include a conclusion section, including future research and limitations of the current research, in addition to the discussion.
Author Response
Authors’ Response to Reviews
The authors would like to thank the reviewers and editor for their valuable comments. The recommended changes and detailed explanations are as follow. All changes in the manuscript are indicated with red color.
Reviewer 3 :
- Authors should present a flowchart or diagram showing the overall research framework or process.
* The figure below illustrates the flowchart of our research.
Figure 1. The flowchart of study.
- Authors should clearly detail the dataset used in tables to facilitate readers' study and research development.
* Thank you for the valuable comment. Informative details about the volunteers participating in the study are provided consecutively in Table 4 and Figure 12 to ensure clarity and facilitate the readers’ understanding.
- Has this research been developed into an application? If so, this should be further presented to complement the presented hardware.
* We would like to note that the system used in the experiments is currently in an experimental stage. However, within the scope of an ongoing project, we will work on developing it into a product suitable for more professional use.
- If the research results can clearly show the sample results obtained from classification, this will further demonstrate the contribution of the research.
* We thank the reviewer for the insightful suggestion. We fully agree that presenting sample classification results would strengthen the contribution of the study. Unfortunately, due to the constraints of the current revision timeline, it is not feasible to include this additional analysis at present. We will, however, take this valuable recommendation into account for future studies and extended versions of this work.
- Authors should include a conclusion section, including future research and limitations of the current research, in addition to the discussion.
* We appreciate the reviewer’s suggestion. In accordance with the comment, a separate Conclusion section has been added, which includes both the limitations of the current research and directions for future research, in addition to the Discussion section.

Reviewer 4 Report
Comments and Suggestions for Authors
Detecting cancer at an early stage is crucial, increasing the likelihood of successful treatment. Detecting human health problems based on the chemical composition of exhaled air is a recognized screening method. Using an artificial nose system for diagnostic purposes is undoubtedly a very good solution, due to its relatively low cost compared to traditional gas analyzers, as well as the simpler measurement procedure. Artificial nose systems are currently being developed in numerous research centers focusing on various areas of application, such as medicine, agriculture, ventilation and air conditioning, detection of hazardous or prohibited substances in public transport, and border protection. Nevertheless, the most important application is in medicine.
For these reasons, I consider the subject of the article to be very important and timely.
Overall, I rate the article highly; it's interesting, the introduction is engaging, and it contains a wealth of information describing the authors' area of interest. The description of the research methods used is extensive, but in my opinion, a bit chaotic, as information about the analysis methods used is mixed with scant information about the results obtained. The "Conclusions" and "Limitations" sections are completely missing. The conclusions from the research are included in the "Discussion" section. I also found no information regarding the electronic devices used for measuring voltage and frequency. It would be interesting to know what software was used to analyze the data.
I also have also a few additional particular comments to the content of the presented article. The order of the comments does not reflect their significance. It results only from the order of appearance in the text of the article.
My remarks and comments:
1. Lines 77-78, “Any imbalance in the functioning of the organs in the body causes the amount of VOCs detected in the breath to change.” - Is there any literature describing such a rule? I have doubts about the term "any" or maybe it should be "most".
2. Lines 90-91, “The identified VOCs were quantified as 33, 54, 34, and 36 for lung cancer, breast cancer, 34 for bowel cancer, and 36 for prostate cancer, respectively [22].” - I suggest correcting this sentence; the quantities 34 and 36 are given twice. This form of the sentence makes it difficult to read.
3. Figure 7 - unexplained designation: a ..e
4. Line 181, “the samples” - I suggest writing "gas samples".
5. Line 189, “a direct current of 12V” - Current is expressed in amperes. The authors most likely meant direct voltage.
6. Line 194, “flow speed” - there should be a flow rate.
7. Line 195, “to directly direct” – I suggest writing “to send directly”.
8. Line 213, “the behavior of the sensor data at the stages of the experiment can be seen”, In the case of MOS sensors, whether the duration of phases II and II is too short is clearly visible. At the end of phase III, most sensors are still fluctuating. I suggest adding some commentary on whether the apparent lack of stability is significant to the measurement and how it was interpreted.
9. Table 1, “Ex-smokers” - shouldn't it be "ex-smokers / non-smokers".
10. Line 251, “where 𝑣𝑛,𝑠,𝑟(𝑡) represents the data for which the value of 𝑣𝑛,𝑠(0) is set to zero” - This statement is incomprehensible to me. In my opinion, it is a change in the reading from the initial value.
11. Line 260, “was converted into conductivity” - so equation 1 is also modified?
12. Line 266, “the same reference point” - What does the reference point mean? Was it one randomly selected sample of exhaled air?
13. Line 268, “breath samples” - there is no information on how the exhaled air samples were collected, how many breaths, and from what place they were taken. Many publications have pointed out the importance of selecting a sampling point.
14. Line 273, “x91” - an explanation is needed where this value comes from.
15. Table 3, in the cell with coordinates LD1: quadrangle 1 and LD2: quadrangle 3, there is "healthy non-smoker", but there should be something else, "healthy smoker" or "LC Patients".
16. Line 381, “on the[” – typo.
17. Figure 17, “calssification” typo
18. Figure 24, I suggest using the same scale on all charts, for example 100%.
Author Response
Authors’ Response to Reviews
The authors would like to thank the reviewers and editor for their valuable comments. The recommended changes and detailed explanations are as follow. All changes in the manuscript are indicated with red color.
Reviewer 4 :
*The "Conclusions" and "Limitations" sections are completely missing. The conclusions from the research are included in the "Discussion" section.
*The Discussion and Conclusion sections have been carefully reviewed and revised accordingly. Necessary improvements were made to ensure clarity, coherence, and alignment with the journal’s standards. The revised sections have been integrated into the manuscript in the appropriate order.
* I also found no information regarding the electronic devices used for measuring voltage and frequency. It would be interesting to know what software was used to analyze the data.
* The data from the MOS sensors were transferred to the computer using a USB-6218 Multifunction I/O Device (National Instruments, Austin, TX, USA). This explanation is added to page 5.
The statement "The data from the QCM sensors were transferred to the computer through a frequency meter circuit integrated with the QCM sensor chamber" is added to page 8. Figure 6 has been updated to clarify the QCM sensor mechanism, and Figure 8 has also been added to the article on page 7 and 8.
Matlab2022b software was used for all operations required for data processing in the study. This statement is also added to page 13.
- Lines 77-78, “Any imbalance in the functioning of the organs in the body causes the amount of VOCs detected in the breath to change.” - Is there any literature describing such a rule? I have doubts about the term "any" or maybe it should be "most".
* Thank you for pointing out this important issue. We agree that the statement “any imbalance in the functioning of the organs” may be too general and not fully supported by the existing literature. Although many studies report that certain physiological and pathological changes in the body can alter the composition of exhaled VOCs, this may not apply to every type of organ dysfunction [1]. To address this concern, we have revised the sentence to adopt a more accurate and cautious formulation. The updated sentence now reads:
“Physiological or pathological changes in organ function can alter the composition of VOCs detected in exhaled breath.”
[1]. Chen, T., Liu, T., Li, T., Zhao, H., & Chen, Q. (2021). Exhaled breath analysis in disease detection. Clinica Chimica Acta, 515, 61-72.
- Lines 90-91, “The identified VOCs were quantified as 33, 54, 34, and 36 for lung cancer, breast cancer, 34 for bowel cancer, and 36 for prostate cancer, respectively [22].” - I suggest correcting this sentence; the quantities 34 and 36 are given twice. This form of the sentence makes it difficult to read.
* The specified sentence was revised and incorporated on page 3 as follows: "As a result of the study, approximately 80% of the VOCs detected in the breath samples of healthy individuals and patients were found to be non-overlapping. Specifically, 34 VOCs were associated with colorectal cancer, 33 with lung cancer, 36 with prostate cancer, and 54 with breast cancer [23]."
- Figure 7 - unexplained designation: a ..e
* In Figure 7, (a)-(d) show the different types of MOS sensors used, while (e) shows the type of QCM sensor used. Also this information has been added in page 8.
- Line 181, “the samples” - I suggest writing "gas samples".
* “The samples” has been edited to “gas samples”
- Line 189, “a direct current of 12V” - Current is expressed in amperes. The authors most likely meant direct voltage.
* The relevant sentence, "A vacuum pump powered by a 12V DC supply was used to deliver breath samples to the sensors," has been revised accordingly and incorporated into the manuscript on page 9.
- Line 194, “flow speed” - there should be a flow rate.
* The flow rate value used in the study is stated as 3.6 L/min immediately after the expression "flow speed" on page 9.
- Line 195, “to directly direct” – I suggest writing “to send directly”.
*“ to directly direct” has been edited to “to send directly” on page 9.
- Line 213, “the behavior of the sensor data at the stages of the experiment can be seen”, In the case of MOS sensors, whether the duration of phases II and II is too short is clearly visible. At the end of phase III, most sensors are still fluctuating. I suggest adding some commentary on whether the apparent lack of stability is significant to the measurement and how it was interpreted.
* The durations of phases II and III were predetermined during the design of the experimental setup. In preliminary trials, longer durations were tested; however, once the sensor responses reached a stable trend, extending the phase time did not yield additional benefits. Therefore, the transition to the next experimental stage was initiated after this stabilization point. The slight decrease in sensor responses observed during Phase III is due to the natural odor washout and sensor recovery processes. At this stage, no new odor molecules were supplied; the sensors were exposed only to the residual volatile compounds in the chamber. As these molecules were gradually adsorbed or dissipated, a decline in the response was expected. The sentence explaining the determination of the duration of the experimental stages has been added to page 9.
- Table 1, “Ex-smokers” - shouldn't it be "ex-smokers / non-smokers".
*“ Ex-smokers” has been edited to “ex-smokers / non-smokers” in Table 4 on page 10 .
- Line 251, “where ??,?,?(?) represents the data for which the value of ??,?(0) is set to zero” - This statement is incomprehensible to me. In my opinion, it is a change in the reading from the initial value.
* We revised the specified section by replacing the expression ?ₙ,ₛ(0) on page 9 with ?ₙ,ₛ(130) to provide greater accuracy. Furthermore, Figures 12 and 13 were added on page 10 to facilitate a clearer understanding of the process.
- Line 260, “was converted into conductivity” - so equation 1 is also modified?
*The data ?ₙ,ₛ,ᵣ(?) obtained from Equation (1) was subsequently used in Equation (3) to derive ?ₙ,ₛ,ᵣ(?), as presented in Figure 14. A comparison of Figures 14 and 16 clearly illustrates the transformation from the initial raw data ?ₙ,ₛ(?) to the conductivity data ?ₙ,ₛ,ᵣ(?) on page 12 and 13.
- Line 266, “the same reference point” - What does the reference point mean? Was it one randomly selected sample of exhaled air?
* In this study, to ensure accurate visualization of the obtained data and to extract meaningful information, reference correction was applied to the data from both MOS and QCM sensors using Equation (1). Accordingly, the ?ₙ,ₛ,ᵣ(?)) data are referred to as reference-corrected data. Both ?ₙ,ₛ,ᵣ(?) and gₙ,ₛ,ᵣ(?) datasets are reference-corrected. By specifying the same reference point, our intention was to emphasize that these two datasets underwent reference correction. To eliminate potential ambiguity, the relevant sentence on page 13 has been revised as follows: “Figure 16 displays an example data from the reference-corrected MOS sensor data and the conductivity of the MOS sensor data.”
- Line 268, “breath samples” - there is no information on how the exhaled air samples were collected, how many breaths, and from what place they were taken. Many publications have pointed out the importance of selecting a sampling point.
*Detailed information about breath sample collection has been added to the breath sample collection section on page 11. Figure 13 has also been added to page 13 to make this process easier to understand. However, since the number of breaths required to fill the TEDLAR bags is not counted when collecting breath samples from patients, clear information on this topic cannot be provided.
- Line 273, “x91” - an explanation is needed where this value comes from.
* Here, 338 denotes the total number of experiments conducted, while 91 corresponds to the total number of extracted features. For instance, considering that there are eight MOS sensors, calculating the maximum value for each sensor results in eight individual feature values, which collectively contribute 8 columns to the feature matrix. By combining the contributions from all feature types employed in this study, the final feature matrix size was obtained as 338 × 91. This paragraph has also been added to page 13.
- Table 3, in the cell with coordinates LD1: quadrangle 1 and LD2: quadrangle 3, there is "healthy non-smoker", but there should be something else, "healthy smoker" or "LC Patients".
* We appreciate the reviewer’s careful examination of the data. In response to the comment regarding the cell with coordinates LD1: quadrangle 1 and LD2: quadrangle 3, we have thoroughly re-checked the dataset and the corresponding classification labels. The label “healthy non-smoker” in this position is correct and consistent with the original subject information. Therefore, no modification was made to this label in the revised manuscript.
- Line 381, “on the[” – typo.
* The phrase “on the[” has been corrected to “on the ” in the revised manuscript.
- Figure 17, “calssification” typo
* The phrase “calssification” has been corrected to “classification” in the revised manuscript.
- Figure 24, I suggest using the same scale on all charts, for example 100%.
All figures in the article have been reviewed, and necessary adjustments have been made to ensure they are at the same scale. However, some figures may have differing scales for better readability and clarity of visuals.

Round 2
Reviewer 2 Report
Comments and Suggestions for Authors
The authors have revised the manuscript and addressed the concerns raised in the comments appropriately. Consequently, I recommend accepting the current version of the manuscript.
Author Response
Response to the Reviewer
We sincerely thank the Reviewer for the positive evaluation of our revised manuscript and for recommending it for acceptance. We are grateful for the constructive comments and suggestions provided during the review process, which significantly helped us improve the clarity and scientific quality of the manuscript.